# Sensory neuron lineage mapping and manipulation in the *Drosophila* olfactory system

Phing Chian Chai[1], Steeve Cruchet[1], Leonore Wigger[2,3] & Richard Benton [1]

Nervous systems exhibit myriad cell types, but understanding how this diversity arises is hampered by the difficulty to visualize and genetically-probe specific lineages, especially at early developmental stages prior to expression of unique molecular markers. Here, we use a genetic immortalization method to analyze the development of sensory neuron lineages in the *Drosophila* olfactory system, from their origin to terminal differentiation. We apply this approach to define a fate map of nearly all olfactory lineages and refine the model of temporal patterns of lineage divisions. Taking advantage of a selective marker for the lineage that gives rise to Or67d pheromone-sensing neurons and a genome-wide transcription factor RNAi screen, we identify the spatial and temporal requirements for Pointed, an ETS family member, in this developmental pathway. Transcriptomic analysis of wild-type and Pointed-depleted olfactory tissue reveals a universal requirement for this factor as a switch-like determinant of fates in these sensory lineages.

[1] Center for Integrative Genomics, Génopode Building, Faculty of Biology and Medicine, University of Lausanne, 1015 Lausanne, Switzerland. [2] Lausanne Genomic Technologies Facility, Génopode Building, Faculty of Biology and Medicine, University of Lausanne, 1015 Lausanne, Switzerland. [3] Vital-IT Group, SIB Swiss Institute of Bioinformatics, 1015 Lausanne, Switzerland. Correspondence and requests for materials should be addressed to R.B. (email: Richard.Benton@unil.ch)

Nervous systems are composed of an enormous number of cell types of diverse structural and functional properties. While the cataloging of cell populations is advancing rapidly through single-cell sequencing approaches[1], the genesis of most cells is poorly understood, limiting our appreciation of the relationships between their developmental trajectories, mature connectivity, and functions. Tracing neuron development from birth to terminal differentiation is a challenge, as this process can occur over a long time period, and across disparate sites within the animal. Direct observation is only practical for numerically simple (and transparent) nervous systems, such as *Caenorhabditis elegans*[2]. In more complex animals, labeling of neural lineages with dyes or retroviruses has been valuable[3–5], though these can lack spatiotemporal resolution and sensitivity. Diverse genetic tools for lineage tracing have been powerful in vertebrates and invertebrates[4–7]. However, a constraint of genetic-based lineage tracing is that unambiguous molecular markers for many populations of neurons either do not exist or are expressed only in late stages of differentiation (e.g. sensory receptor genes).

Here we address the problem of lineage tracing and manipulation in the peripheral olfactory system of *Drosophila*. Adult olfactory organs originate from the larval antennal imaginal discs. Within this tissue, sensory organ precursor (SOP) cells give rise to short lineages producing up to four olfactory sensory neurons (OSNs) and four support cells, which together form a sensillum on the antenna[8–10]. There are ~50 classes of OSNs, defined by their expression of a specific Odorant Receptor (Or) or Ionotropic Receptor (Ir)[11,12]. Stereotyped combinations of 1–4 OSN types are housed within individual sensilla, which belong to several morphological subclasses: antennal basiconic/trichoid/intermediate (ab, at, ai) sensilla (containing Or-expressing OSNs) and antennal coeloconic (ac) sensilla (housing predominantly Ir-expressing OSNs)[8,13,14]. The axons of OSNs expressing the same receptor converge onto a discrete glomerulus within the primary olfactory center (antennal lobe) in the brain[13–16]. Here, they synapse with projection neurons (PNs)—which arise from lineages of central brain neuroblasts[17,18]—that carry olfactory signals to higher brain regions[9,19].

Despite the extensive knowledge of the anatomy and function of the olfactory system[11,19,20], the developmental origins of distinct olfactory SOPs and how these relate to olfactory circuit organization are poorly understood[9,10]. This is largely because of the lack of markers to trace a specific lineage from SOP birth to OSN maturation. Here we applied an immortalization labeling system for OSN lineages, which uses the CONVERT (or FLEX-AMP) technique[21,22] and takes advantage of the large resources of *Drosophila* enhancer-GAL4 driver lines for genetic marking of cell subpopulations[23,24]. This approach permits us to, first, generate an olfactory fate map in the antennal disc, second, visualize an entire olfactory sensory lineage and, third, characterize the role of a novel molecular determinant of OSN development.

## Results

**An immortalization labeling system for OSN lineages**. We immortalized the expression of antennal disc-expressed GAL4 drivers within a time window spanning SOP specification through three events (Fig. 1a): (i) temporally controlled heat-inactivation of GAL80$^{ts}$ (a thermosensitive inhibitor of GAL4), (ii) GAL4 induction of Flippase-mediated recombination and activation of a LexA driver, (iii) LexA-dependent expression of a Green Fluorescent Protein (GFP) reporter in the labeled SOPs and their descendants.

To assess labeling efficiency, we first compared GFP signals from nonimmortalized and immortalized GAL4 drivers for the antennal proneural genes, *atonal* (*ato*) and *absent MD neurons*

*and olfactory sensilla* (*amos*), which are expressed in SOPs that form ac and ab/at/ai sensilla, respectively[25,26]. In antennal discs, we focused on the proximal region (A3), labeled by the patterning determinant Dachshund (Dac)[27], where olfactory SOPs are located (Fig. 1b). In adults, we identified GFP-labeled OSNs based upon their glomerular innervations (Fig. 1c).

At 4 h after puparium formation (APF), when SOP specification is occurring (Fig. 1a), *ato-GAL4* labels many SOPs (Fig. 1d). As *ato* expression is downregulated by 12 h APF (prior to SOP division and neuron differentiation)[25], the nonimmortalized driver does not label any OSNs (Fig. 1d, e). By contrast, immortalized *ato-GAL4* labels OSNs in all *ato*-dependent sensilla (i.e. ac1−4, as well as those in the antennal sacculus and another olfactory organ, the maxillary palp) (Fig. 1d, e). Similarly, nonimmortalized *amos-GAL4* was detected only in the disc, but when immortalized, labels all OSNs from ab, at, and ai sensilla (Fig. 1d, e).

We next tested drivers for three olfactory coreceptor genes (*Ir8a-GAL4*, *Ir25a-GAL4* and *Orco-GAL4*), which are only expressed during later stages of neuron differentiation in many populations of OSNs (Fig. 1d, e)[28,29]. As expected, when these drivers were immortalized during early SOP development, no OSN GFP labeling was observed (Fig. 1d, e).

Finally, we asked if the immortalization system could capture driver expression that is unstable/transient. We chose *engrailed* (*en*)-*GAL4* because the expression of *en* is highly dynamic at early pupal stages (up to 9 h APF) before stabilizing in progenitor cells[30]. *en-GAL4* is expressed in a large zone of the antennal disc at 2 h APF, but is restricted to just 16 OSN classes in the adult (Fig. 1d, e). We immortalized this driver in either early (4 h before puparium formation (BPF)-20 h APF) or late (9–39 h APF) time windows. Early immortalization led to GFP labeling of most OSN classes, consistent with the extensive *en* expression in early pupae (Fig. 1d, e). Late immortalization restricted labeling to fewer glomeruli, approaching the number labeled by the nonimmortalized driver, suggesting this time window reflects *en* expression once it has largely stabilized into the terminal adult pattern (Fig. 1d, e).

Together, these results indicate that the immortalization strategy effectively captures and preserves GAL4 driver expression during a desired developmental time window to relate early expression patterns in disc SOPs to the OSN lineages that arise from these precursors. There is no nonspecific labeling of OSNs without the immortalization (heat-inactivation) step or in the absence of an *enhancer*-GAL4 (Supplementary Fig. 1a, b).

**A fate map of olfactory sensory organ precursors**. A previous analysis[31] proposed a fate map of SOPs in the antennal disc (i.e. defining the spatial origins of different OSN classes). This map was created, in part, by relating the endogenous expression of various patterning factors in the disc to the OSNs labeled by transgenic drivers for these same factors in later stages. Such an approach assumes that these patterning factors do not change their expression during SOP development. However, the observation that drivers for patterning factors often label only a subset of OSNs within the same sensillum—which derive from the same SOP[31]—indicates that this assumption is incorrect.

To explore this issue further, we re-examined the expression of two previously used drivers, *Bar-GAL4* and *apterous* (*ap*)-*GAL4*. In the antennal disc, both drivers are expressed in the presumptive arista (PA) zone (Fig. 1b, d). Previous analyses showed that in adults *Bar-GAL4* labels three OSN populations (VA1d/Or88a, VL2a/Ir84a, and VL1/Ir75d) while *ap-GAL4* labels six populations (DA3/Or23a, VA1d/Or88a, DL3/Or65a/b/c, DM4/Or59b, DL5/Or7a, VM2/Or43b, and VL2p/Ir31a) in pupae (but not adults)[31]. These observations, together with loss- and

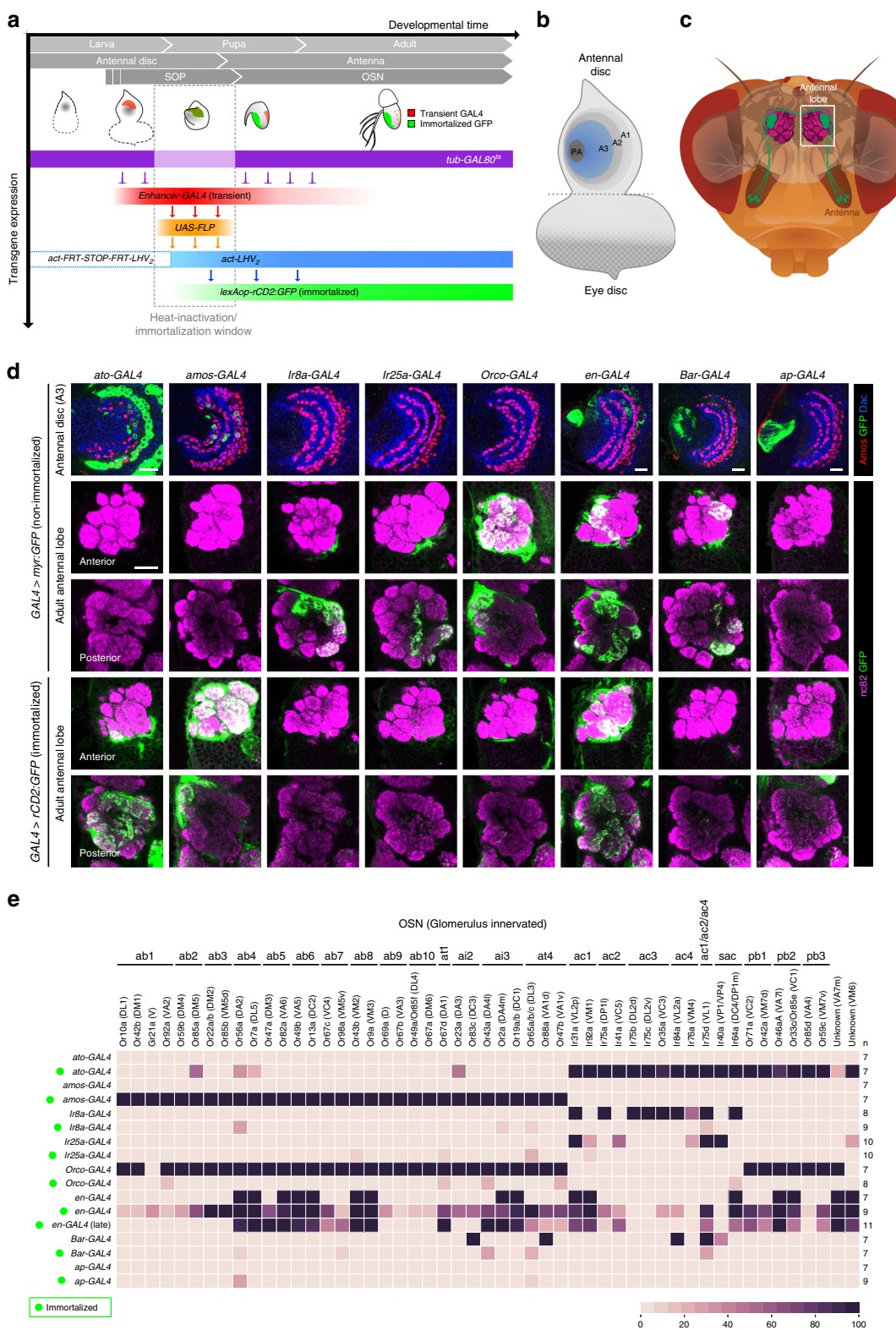

gain-of-function analyses, led to the proposition that SOPs for the corresponding sensilla (ab2, ab4, ab8, at2, at4, ac1, ac2, and ac4) all lie within the PA[31]. By contrast, we found that neither immortalized *Bar-GAL4* nor immortalized *ap-GAL4* consistently label any antennal lobe glomeruli (Fig. 1d, e), and the only GFP-positive neurons we detect are located in the arista (Supplementary Fig. 1c, d). As previously proposed[32], the PA contains SOPs only for the aristal thermosensory neurons, but not olfactory

sensilla. These observations argue that the reported OSN expression of the nonimmortalized *GAL4* drivers reflects late expression patterns unrelated to their disc expression (Supplementary Fig. 1e) and/or labeling of nonantennal neurons (Supplementary Fig. 1f).

With our immortalization approach, we set out to reconstruct a new olfactory SOP fate map that would align with findings from previous studies of SOP development[25,26,33–35]. We first

**Fig. 1** A genetic immortalization labeling system for OSN lineages. **a** Schematic of *Drosophila* peripheral olfactory system development and the genetic immortalization strategy. **b** Schematic of the larval eye-antennal imaginal disc; olfactory SOPs develop in the A3 region (blue). PA presumptive arista zone. **c** Schematic of the *Drosophila* head, illustrating a single population of OSNs expressing the same olfactory receptor (green); these project axons from the antenna at the periphery towards a unique glomerulus in the antennal lobe in the brain (dashed box). **d** Row 1: nonimmortalized *enhancer*-GAL4-driven expression of myr:GFP (green) in 2 h APF antennal discs (except for *ato*-GAL4 and *amos*-GAL4, which show 4 h APF discs, due to the delayed expression of these drivers). The discs are colabeled with α-Amos (red) and α-Dac (blue) to provide spatial landmarks. Rows 2–3: nonimmortalized *enhancer*-GAL4-driven expression of myr:GFP (green) in the axon termini of OSNs innervating the right antennal lobe of the adult. The glomerular structure of the lobe is visualized with α-Bruchpilot (nc82, magenta); anterior and posterior focal planes (single optical slices) are presented to show the majority of glomeruli. Rows 4–5: immortalized *enhancer*-GAL4-driven expression of rCD2:GFP (green) in OSN axons, presented as in rows 2–3. The immortalization time window was 4 h BPF-20 h APF, during which the SOPs undergo maturation (Fig. 1a). Note that the immortalized *enhancer*-GAL4s (rows 4–5) label different subsets of OSNs from their nonimmortalized counterparts (rows 2–3). Scale bars = 20 μm. The genotypes of the flies presented in the figures are listed in Supplementary Table 1. **e** Heatmap showing the frequency that a given antennal lobe glomerulus is innervated by OSN axons labeled by a GFP reporter (myr:GFP or rCD2:GFP) under the control of a nonimmortalized or immortalized *enhancer*-GAL4 driver. sac sacculus sensilla, pb maxillary palp basiconic sensilla; see text for other abbreviations. The number of antennal lobes scored (*n*) is indicated on the right. Immortalization was performed during the same time window (4 h BPF-20 h APF) for all drivers except for late *en*-GAL4 immortalization (9 h APF-39 h APF)

partitioned the A3 domain of the antennal disc into eight concentric arcs (1–8), using Amos and Dac to delineate their boundaries (Fig. 2a). Dac defines a broad, curved zone in A3 with strong central expression and weaker expression at both dorsal and ventral edges. Amos is expressed in SOPs that form three parallel arcs[34]. At 2 h APF, the SOPs within the two inner arcs (2 and 4) coexpress high levels of Dac; the wider outermost arc overlapped with the strong-to-weak transition border of Dac expression, allowing us to subdivide it into two adjacent arcs (6 and 7). The Dac-positive, Amos-negative regions constitute the remaining four arcs (1, 3, 5, and 8) (Fig. 2a). These regions are presumably occupied by Ato-positive SOPs because they flank and intercalate with Amos-expressing SOPs (Fig. 2b). To increase the resolution of the spatial map, we divided each arc along the depth of the antennal disc arbitrarily into superficial and deep layers (Fig. 2c).

To genetically label different SOP subsets, we screened images of >500 antennal disc GAL4 drivers[24], and identified 25 that exhibited restricted expression in the A3 region. Of these, 19 were eliminated as their preliminary immortalization analysis produced inconsistent expression patterns, suggesting these had unstable disc expression (see Methods). The expression of the remaining six lines, along with a driver for *rotund* (*rn*) (a patterning factor with stable expression[31,36]), was examined within the spatially partitioned A3 domain at 2 h APF (Fig. 2d and Supplementary Fig. 2a). Despite the limited number of lines, each arc on the map could be uniquely identified based on the combinatorial expression patterns of these drivers, as well as the known expression domains of Amos and Ato (Fig. 2f). Superficial and deep layers were only resolved for arcs 2 and 4 (Fig. 2f).

We next immortalized these seven GAL4 drivers and examined the identities of the labeled glomeruli (Fig. 2d, e). Antennal deafferentation experiments confirmed that the specific glomerular signals were entirely due to the contribution of OSNs (Supplementary Fig. 2b). While we assume that most of these OSN classes originate from SOPs within the corresponding GAL4 expression zone observed at 2 h APF, we cannot definitively exclude that some OSNs might originate from other, out-of-zone SOPs that express GAL4 at earlier or later time points during the immortalization window (4 h BPF-20 h APF). However, given the stochastic nature of the labeling method, we suspected such out-of-zone SOPs would lead to less frequent GFP labeling than those SOPs that robustly express the driver at 2 APF. By comparing the GAL4 expression patterns on the SOP spatial map and GFP-labeled OSNs after immortalization, we were able to assign the SOPs for at least 16/18 antennal olfactory sensillar classes to specific arcs (Fig. 2f). For example, all OSN classes within ab1,

ab9, and at4 sensilla are labeled at high frequency with the immortalized *GMR13B04-GAL4* and *GMR24C12-GAL4*, allowing us to infer that SOPs for these sensilla lie within arc 6, as this is the only Amos-positive arc where only these drivers are expressed (Fig. 2e, f). The assignment of at4 to this arc is less definitive, because OSNs in this sensillum type are also labeled, albeit much less frequently, by other immortalized GAL4 drivers; this may reflect transient expression of these drivers in this arc during the immortalization window.

**Neuroanatomical correlates of the SOP fate map**. We asked whether the position of SOPs on the array of antennal disc arcs has any relationship with the properties of the corresponding circuits in the adult. In contrast to the earlier fate map[31], we find that individual subtypes of ac, ai, and at sensilla originate from distinct arcs (Fig. 2f); for the more numerous ab sensilla, several subtypes derive from the same arc (Fig. 2f). Mapping of arc identity onto the spatial distribution of sensilla in the adult antenna[37] revealed partial preservation of the concentric organization of different lineages (Fig. 3a). There are, however, some exceptions (e.g. a lateral subset of ac3 sensilla), which may reflect both the distortion of the original spatial relationships of SOP lineages due to eversion of the antennal disc during metamorphosis, as well as the local dispersion of sensilla in the pupal antenna[30]. At the molecular level, we observed that individual arcs gives rise to OSNs expressing Ors (or Irs) that are not necessarily confined to a specific clade on phylogenetic trees of these receptor families (Fig. 3b).

Examination of arc identity and antennal lobe glomerular organization revealed a clear segregation of OSNs derived from Amos- and Ato-expressing arcs, as previously described[13,14] (Fig. 3c). However, we only detected significant, albeit mild, clustering of glomeruli innervated by OSNs derived from a subset of arcs (2, 5 and 7) (Fig. 3c).

Finally, we noted a bias in SOP arc placement and the birth order of the corresponding PN synaptic partners in the anterodorsal PN neuroblast lineage[18]: of the early (embryonic) born PNs the majority (10/13) partner with OSNs originating from more internal arcs (1–3), while of the late (larval) born PNs the majority (12/13) partner with OSNs originating from more peripheral arcs (6–8) (Fig. 3d). We speculate that this pattern reflects a remnant of evolution, and propose the following model: putative ancestral olfactory circuits (as deemed by their expression of more deeply conserved Irs[38]) are formed from OSNs derived from internal arcs and early-born PNs. More recently evolved olfactory pathways (expressing Ors of more limited

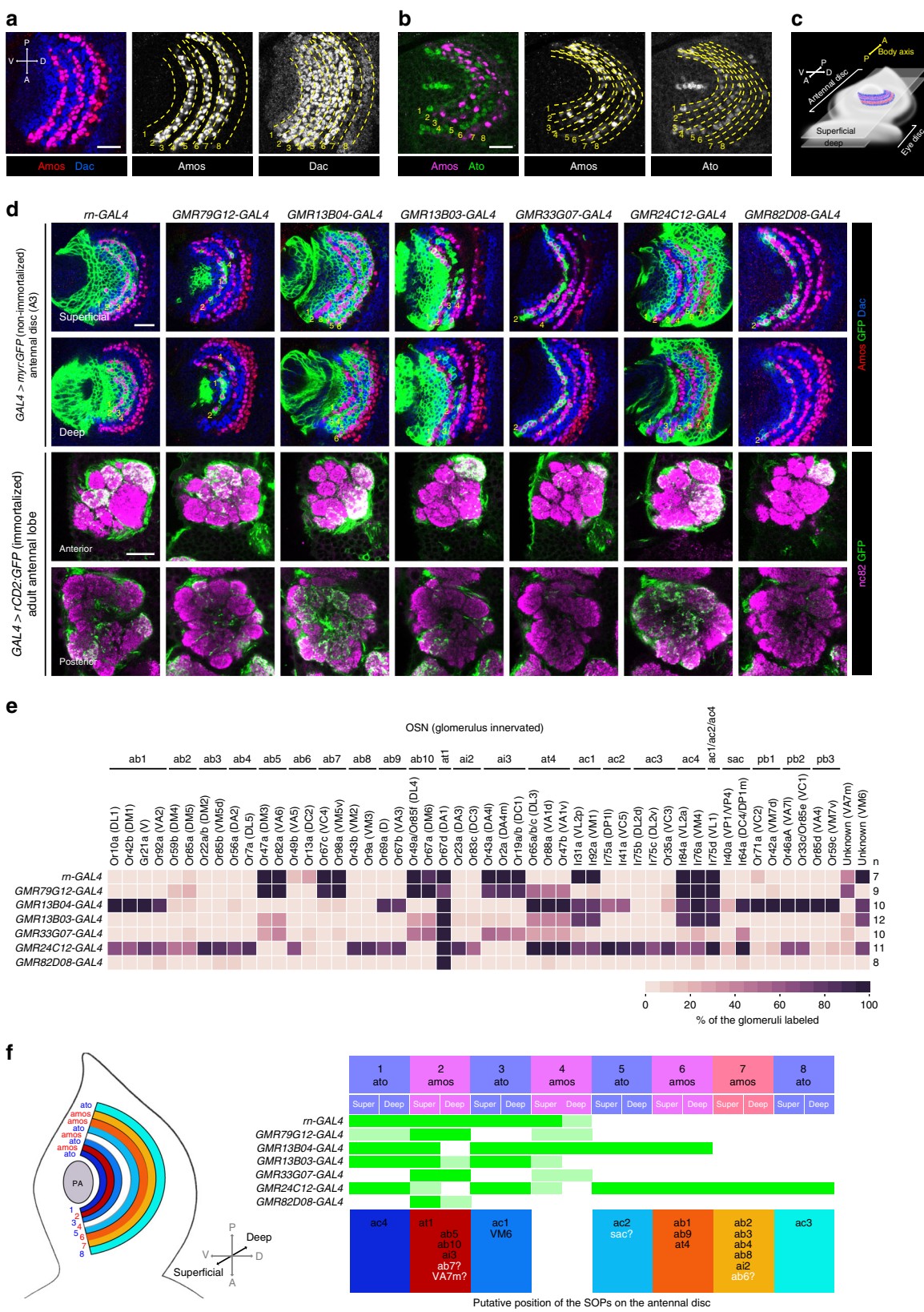

phylogenetic conservation) comprise OSNs derived from SOPs in arcs that were added externally to the array in the disc, and PNs that were generated by extension of the central neuroblast divisions.

**A genetic driver for a single OSN lineage.** Among the lines used to generate the fate map, we further focused on *GMR82D08-GAL4*. This driver labels SOPs that exclusively produce at1 sensilla (Fig. 2d, e), which contain a single, Or67d-expressing OSN that

**Fig. 2** An olfactory SOP fate map. **a** Immunolabeling of a 2 h APF antennal disc (A3 region) with α-Amos (red) and α-Dac (blue), whose expression boundaries allow the definition of eight distinct concentric arcs. Scale bar = 20 μm (in this and other panels). **b** Immunolabeling of a 4 h APF antennal disc with α-Amos (magenta) and α-Ato (green) showing that arcs 1, 3, 5, and 8 are occupied by SOPs for ac sensilla (Ato-positive) while arcs 2, 4, 6, and 7 are occupied by SOPs for ab, ai, and at sensilla (Amos-positive). **c** Schematic showing the location of the arcs (indicated by the SOPs with the same color profile as in (**a**)) on the antennal disc to illustrate the position of the superficial and deep layers. **d** Rows 1–2: nonimmortalized *enhancer*-GAL4-driven expression of myr:GFP (green) in 2 h APF antennal discs. The discs are colabeled with α-Amos (red) and α-Dac (blue) to reveal GFP expression relative to the concentric arcs (yellow numbers). Images of superficial and deep layers of the disc are shown. Rows 3–4: immortalized *enhancer*-GAL4-driven expression of rCD2:GFP (green) in the axon termini of OSNs innervating the antennal lobe of the adult (presented as in Fig. 1d). The immortalization time window was 4 h BPF-20 h APF. **e** Heatmap showing the frequency that a given antennal lobe glomerulus is innervated by OSN axons labeled by rCD2:GFP under the control of the immortalized *enhancer*-GAL4 drivers shown in (**d**). The number of antennal lobes analyzed (*n*) is indicated on the right. **f** Left: schematic of the antennal disc showing the Ato- and Amos-positive arcs. Right: an olfactory fate map, summarizing the expression pattern in different arcs of the *enhancer*-GAL4 lines (green: strong expression/complete coverage; light green: weak expression/incomplete coverage). The birthplace of SOPs of different OSN/sensillar classes (bottom row) was deduced from the similarity of the expression pattern of these lines in the antennal disc and the similarity of their OSN labeling patterns when immortalized during SOP development. Black labels: high certainty of SOP assignment; white letters with question mark: ambiguous assignment

projects to DA1[39,40]. This line (hereafter, at1 driver) was of interest in permitting developmental analysis of a specific OSN lineage. The at1 driver is relatively stably expressed in the SOPs located around arc 2 of the antennal disc from late larval until early pupal stages (Fig. 4a). From 20 APF, certain cells within the newly formed SOP-derived cell clusters start downregulating at1 driver expression until it becomes undetectable by 48 h APF (Fig. 4a). When immortalized, the at1 driver persists to label ∼35 cell clusters (mean 35.4 ± 6.4 SD clusters; *n* = 15 antennae) in the adult, of which ∼33 contain a single cell that expresses *Or67d* mRNA (33.1 ± 4.1/35.4 ± 6.4 (93.5%) clusters) (Fig. 4b, c). The remaining four cells in the labeled clusters correspond to sensilla support cells, as they express the odorant binding protein Lush[41] (Fig. 4d). As there are ∼60 Or67d OSNs in the adult antenna[37], these observations indicate that the at1 driver is expressed specifically, though incompletely, in at1 SOPs.

**Analysis of OSN lineage properties with the at1 driver.** Previous analysis of olfactory lineages, using random labeling approaches and visualization of several transcription factors, allowed reconstruction of a model of SOP lineage divisions from observation of many independent clones[35,42]. An important drawback of this approach is its lack of specificity for visualizing particular SOP subtypes. Indeed, we found that clones generated in this manner within the same time window can be variable both in cell number and expression of developmental markers (Supplementary Fig. 3a, b). It is impossible to distinguish whether this is because they represent distinct SOP lineages or because SOPs of the same type develop asynchronously. Using our immortalized at1 driver to visualize this single olfactory lineage, we observed that the timing of cell division and transcription factor expression in different at1 SOPs is largely, though not perfectly, synchronized (Fig. 5a). This observation indicates that the random clonal approach visualizes, as expected, multiple lineages, which confounds appreciation of the precise developmental properties of a specific lineage.

We therefore reanalyzed the spatiotemporal properties of molecular marker expression throughout the at1 lineage. Observation of just two lineage markers, Senseless (Sens) and Seven-up (Svp), allowed us to establish the precise birth order of cells, which was not possible in the random labeling approach[35,42] (Fig. 5b). Visualization of additional markers (Partner of Numb (Pon), Hamlet (Ham), Embryonic lethal abnormal vision (Elav), Cut (Ct), and Prospero (Pros)) permitted unambiguous determination of the identities of all intermediate and terminal daughter cell fates of this lineage (Fig. 5b, c and Supplementary Fig. 3c–e). These comprise the Nab cell (which gives rise to the Or67d OSN[42]), the three other cells of the neuronal lineage that undergo

apoptosis (Naa, Nba, Nbb), and four support cells (Oaa, Oab, Oba, Obb).

**A screen for molecules controlling at1 lineage development.** Although several factors that function in OSN lineage-specification have been described[10], our knowledge of this process—encompassing the determination and coordination of olfactory receptor expression with axon guidance to a unique glomerulus—is incomplete. To identify novel molecules regulating Or67d OSN specification, we conducted a transgenic RNAi screen of 808 genes encoding candidate transcription factors, chromatin regulators and embryonic patterning genes (see Methods). Knockdown of 121 genes showed various forms of antennal defects; secondary screening with independent transgenes yielded 35 high confidence hits that gave reproducible phenotypes with at least two RNAi constructs (Fig. 6a).

We performed deeper analysis of these 35 genes by inducing RNAi using two GAL4 drivers with different spatiotemporal expression profiles: (i) a constitutive, eye-antennal disc-specific driver (*ey*-FLP, *act*-GAL4 flip-out cassette), which is active from the second instar larval stage onwards (Supplementary Fig. 4), and (ii) a late sensory neuron driver (*pebbled* (*peb*)-GAL4), which is activated during OSN differentiation[43] (Fig. 6b). To focus on Or67d OSN fate specification, we performed RNAi in flies expressing an Or67d:GFP reporter (this additionally labels Or82a OSNs due to ectopic transgene expression)[13]. To inform both OSN fate specification and wiring, we visualized the innervation of neurons expressing this reporter in the antennal lobe, and classified the RNAi phenotypes into distinct categories (Fig. 6c). While all of these genes gave strong phenotypes with the constitutive driver, only 12/35 did so with the late driver, implying that the majority is important for early developmental processes (Fig. 6d). Loss of function of most of these genes led to a reduction of Or67d axon density in DA1, potentially caused by misspecification in the at1 lineage of OSN fate, receptor expression and/or OSN targeting (Fig. 6c, d). One intriguing exception was observed in *pointed* (*pnt*) RNAi flies, which exhibited a higher density of Or67d OSNs innervating the antennal lobe, as revealed by an enlarged DA1 glomerulus (Fig. 6c). A similar phenotype was observed for Or82a OSNs that target the VA6 glomerulus (Fig. 6c).

To understand the basis of the enlarged glomerulus phenotype, we examined reporter expression in the antenna and observed a substantial increase in the number of GFP-positive OSNs (81.2 ± 16.9 (*n* = 12 antennae) in control animals and 177.5 ± 33.3 (*n* = 10 antennae) in *pnt* RNAi animals). Moreover, while the soma of the labeled OSNs in control antennae are separated from one another (singlets), reflecting their compartmentalization in

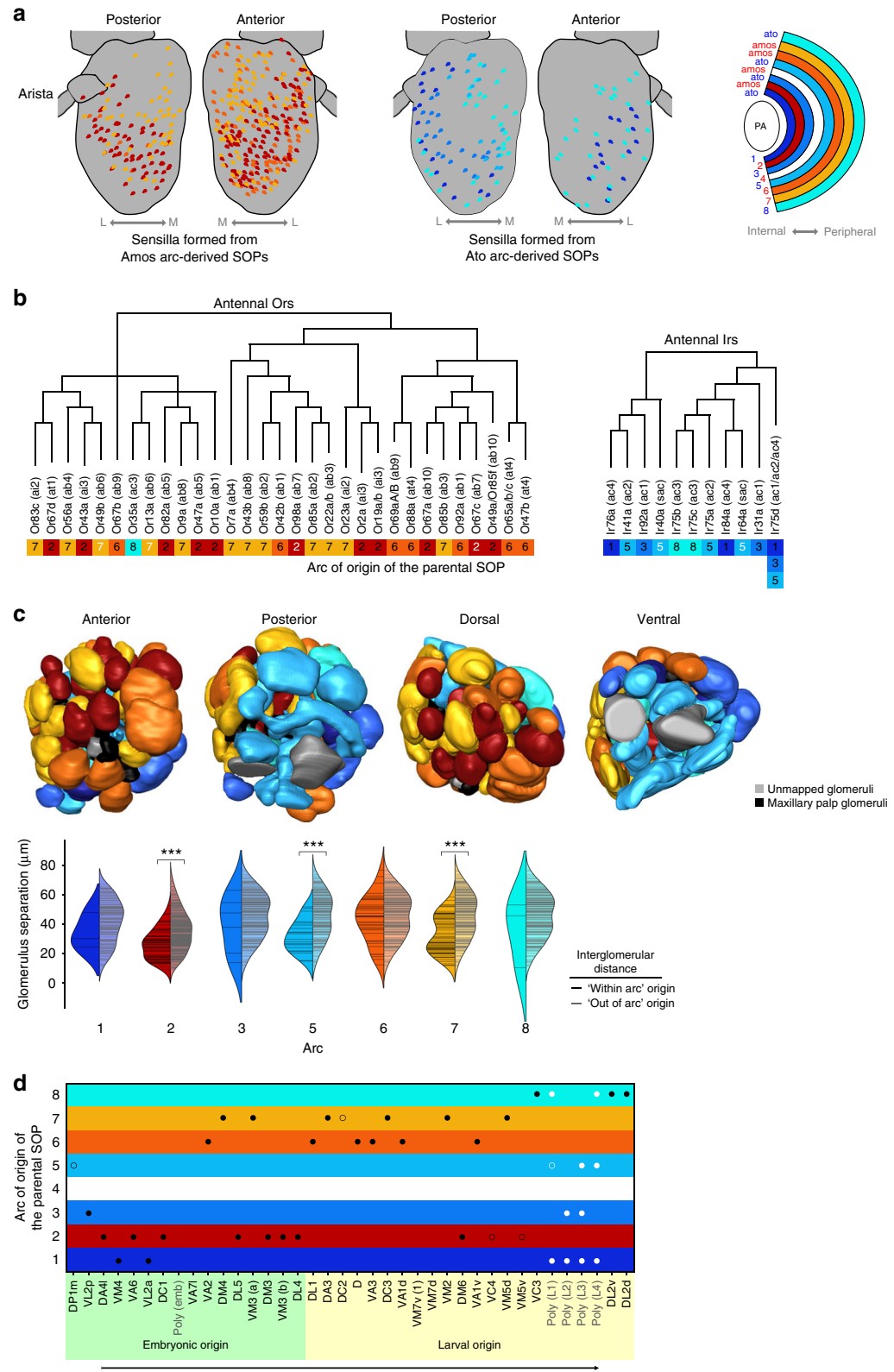

distinct sensilla, in *pnt* RNAi antennae most of the GFP-expressing soma occur in clusters of two cells (doublets) (Fig. 7a). This phenotype was not observed using the late driver of *pnt* RNAi (Figs 6d and 7a), suggesting that *pnt* regulates OSN fate prior to terminal OSN differentiation. Importantly, we found that the at1 driver is also capable of inducing *pnt* RNAi to produce Or67d neuron duplication (Fig. 7b).

The duplication of Or67d neurons in *pnt* RNAi antennae observed with the GFP reporter was validated by direct visualization of *Or67d* mRNA expression (Fig. 7c). Furthermore, we confirmed this RNAi phenotype through analysis of a *pnt* loss-of-function mutation, by using MARCM[44] to generate GFP-labeled wild-type or *pnt* mutant OSN clones, which were examined for *Or67d* mRNA expression.

**Fig. 3** Correlations between the olfactory fate map and circuit properties. **a** Distribution of different sensillar classes (from ref. [37]), color-coded for their antennal disc arc of origin (one color may represent many different sensilla types that arise from the same arc). Sensilla originating from both Amos-positive arcs (red-yellow gradient) and Ato-positive arcs (blue-cyan gradient) are shown separately for clarity; both partially preserve their relative positions in the antenna: the internal arcs produce sensilla that have lateral (L) distributions while the peripheral arcs produce sensilla with more medial (M) distributions. **b** The relationship between Or/Ir protein phylogeny and the arc of origin of the corresponding SOPs (derived from the sensilla fate map in Fig. 2f). The Or phylogenetic tree is adapted from ref. [70]; the Ir phylogenetic tree is adapted from ref. [14]. The arcs from which the OSNs originate are color-coded as in (**a**). Arc labels in black: high confidence OSN/sensillum mapping; arc labels in white: ambiguous OSN/sensillum mapping. **c** Different 3D surface views of the right antennal lobe (from ref. [16]), in which glomeruli are colored according to the arc of origin of the SOPs from which the corresponding OSNs arise (as in (**a**)). The split violin plots compare the pairwise distances between glomeruli[71] associated with a particular arc (denoted by black horizontal bars within the left, darker-colored kernel density estimation area), and the pairwise distances between each of the individual glomeruli associated with a particular arc and all other glomeruli not associated with the arc (gray bars within the right, paler-colored kernel density estimation area). Significant differences in distributions based upon one-sided Mann−Whitney $U$ test are indicated: ***$p < 0.0001$. **d** Relationship between OSN birthplace and PN birth order. OSN classes (defined by their glomerular innervation) are positioned on the vertical axis according to the arc of origin of the SOPs from which they arise, and ordered on the horizontal axis by the birth timing of their synaptic partners produced in the anterodorsal PN (adPN) neuroblast lineage (based on data from ref. [18]). Black/white circles indicate OSNs mapped to mono-/polyglomerular PNs, respectively (filled circles: OSNs from sensilla mapped with high certainty; unfilled circles: OSNs from ambiguously mapped sensilla)

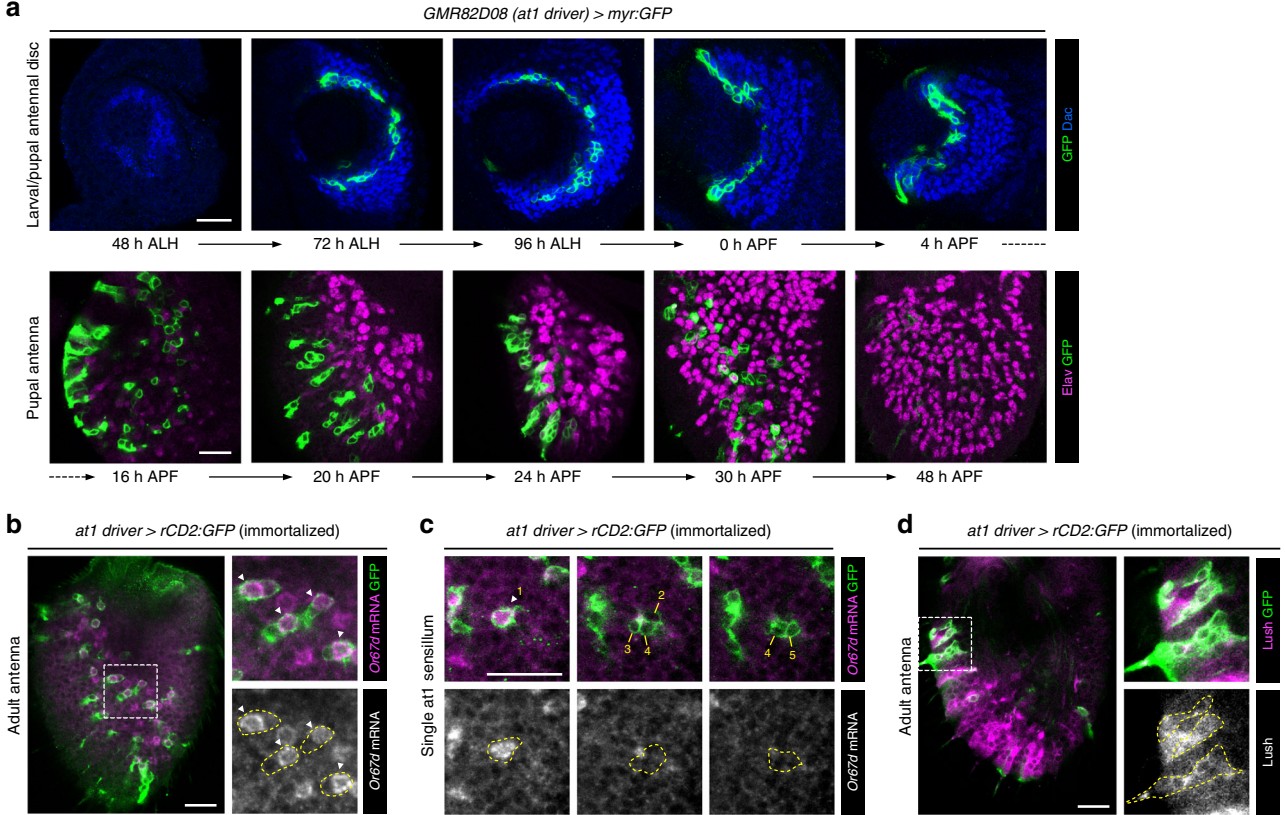

**Fig. 4** An OSN lineage-specific driver. **a** Top row: developmental expression of the nonimmortalized *GMR82D08-GAL4* (hereafter, at1 driver) using a myr:GFP reporter (green) in the antennal disc SOPs (region marked by α-Dac (blue)) during late larval/early pupal stages. Bottom row: the at1 driver is expressed in the daughter cells of these SOPs in the developing pupal antenna but progressively loses its expression from 20 h APF as OSNs differentiate (visualized with the neuronal marker α-Elav (magenta)). Scale bar = 20 μm in this and other panels. **b** Immortalization of the at1 driver reveals labeling of clusters of cells in the adult antenna by an rCD2:GFP reporter (green). RNA fluorescence in situ hybridization demonstrates that a single cell within each cluster (arrowheads in the inset images) expresses *Or67d* mRNA (magenta). **c** Representative example of a single sensillum in the adult antenna labeled by the immortalized at1 driver, viewed at three focal planes. There is a single *Or67d* mRNA-positive OSN (cell 1, arrowhead), flanked by four non-neuronal support cells (cells 2–5). **d** Sensilla cells labeled by the immortalized at1 driver lineage (α-GFP; green) also express Lush (magenta), an odorant binding protein unique to trichoid sensilla support cells[72]

All *Or67d*-expressing neurons in control clones were singlets; by contrast, all *pnt* clones contained doublets of *Or67d* mRNA-positive neurons (Fig. 7d).

**Pnt is expressed dynamically within the at1 lineage.** To understand the function of Pnt in the at1 lineage, we first examined the spatiotemporal expression pattern of this transcription factor. Because Pnt is expressed in many different cells in the pupal antenna (Fig. 8a), our immortalized at1 driver line was essential to follow its dynamic expression explicitly in this lineage (Fig. 8a, b). Pnt is weakly, but broadly, expressed at early pupal stages (2 h APF) (Supplementary Fig. 5a) and becomes

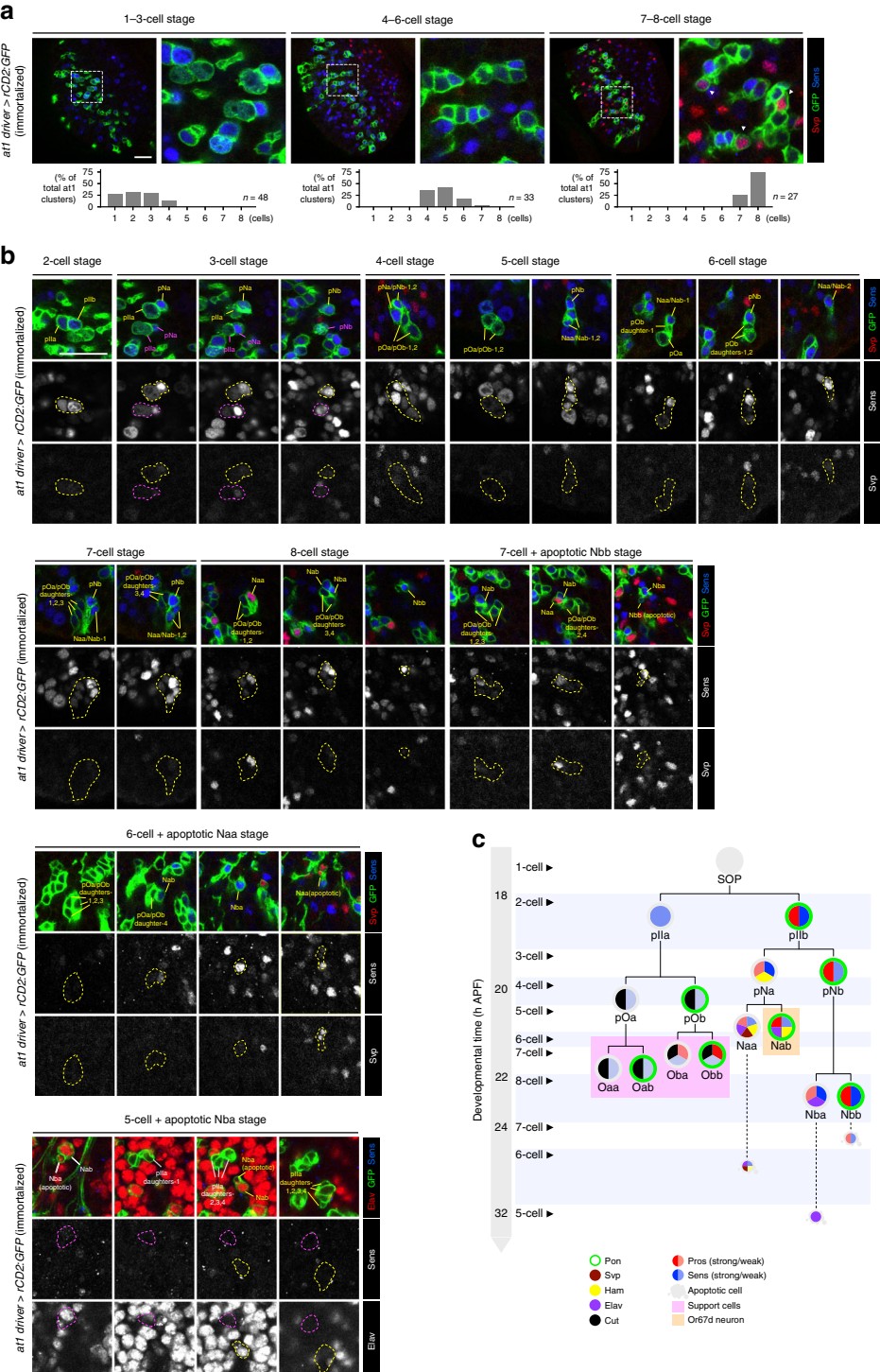

**Fig. 5** The at1 lineage is synchronized in cell division sequence and marker expression. **a** Full views of the immortalized at1 driver-labeled lineages (using an rCD2:GFP reporter; green) on pupal antennae at three developmental time points. For each time point, all the at1 lineages have comparable numbers of daughter cells and identical Svp (red) and Sens (blue) expression patterns (shown in the magnified views of the boxed regions). The histograms show the percentage of at1 lineages at different cell stages. Scale bar = 20 μm in this and other panels. **b** Close-up views of the SOP and its daughter cells in the at1 lineage (green) reveal a highly stereotyped series of cell divisions from 2 to 8 cells, before the Nbb, Naa and Nba cells undergo sequential apoptosis. Most of the intermediate and terminal daughter cells in this lineage could be unambiguously identified based on the expression of Svp (red) and Sens (blue), as well as their position within the cluster (division typically occurs along the same axis). For the last apoptotic Nba stage, Elav (red) marks the neural-specific Nab and Nba cells. Where necessary, multiple focal planes are shown to visualize all the cells in each cluster. The yellow or magenta dashed lines within each cell stage outline the same at1 cluster at different focal planes (across multiple panels). **c** Schematic showing the division time, birth sequence and expression pattern of molecular markers in the at1 lineage (based also upon data from Supplementary Fig. 3c–e)

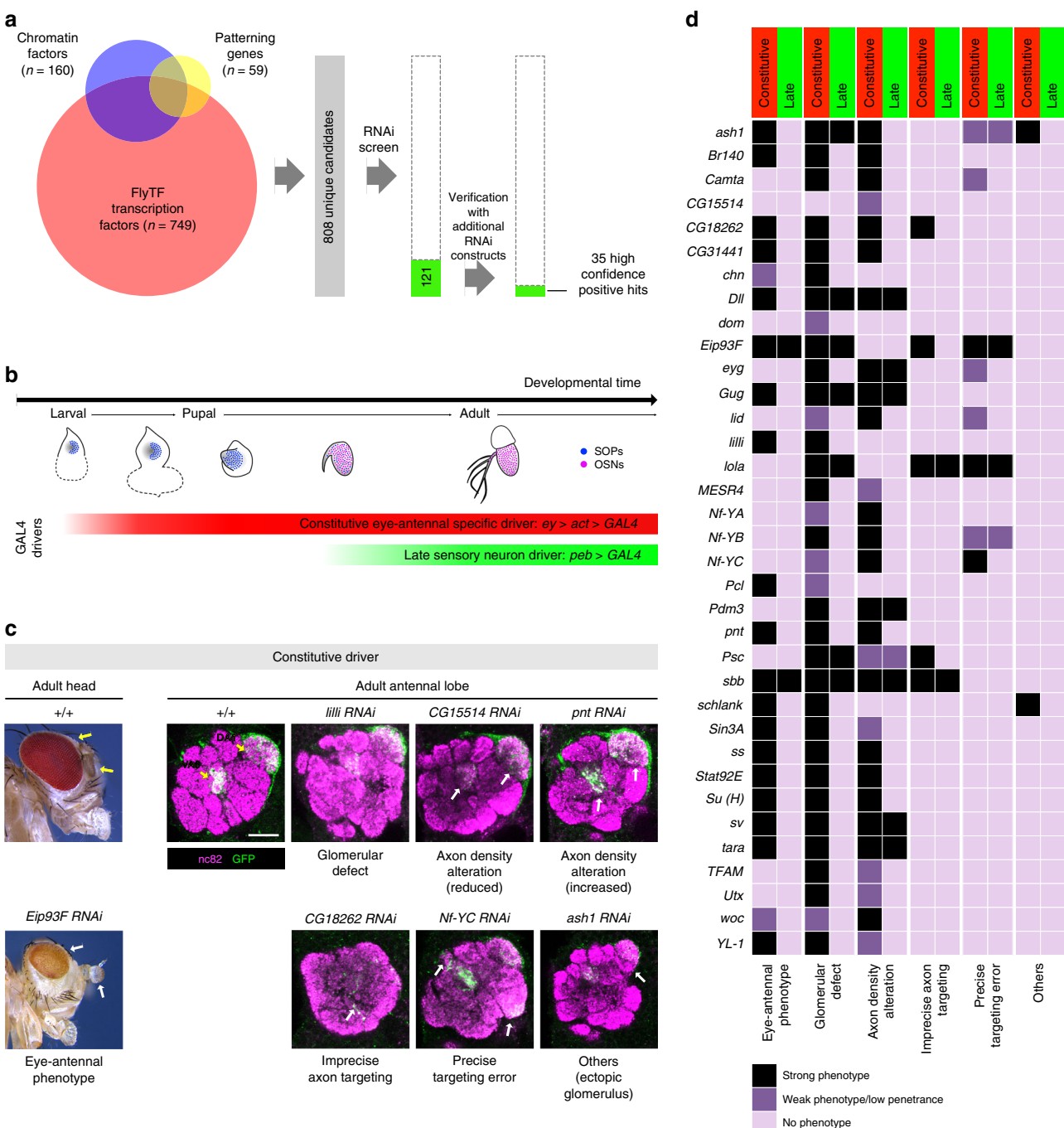

**Fig. 6** An RNAi screen for molecular determinants of at1 lineage development. **a** RNAi screen workflow: two rounds of screening, starting from 808 candidate genes (see Methods) identified 35 genes that showed reproducible defects in antennal development. **b** Temporal expression of GAL4 driver lines used for RNAi: the constitutive driver, comprising *ey-FLP* and an *actin-GAL4* flip-out cassette, is expressed specifically in the eye-antennal tissue from the second instar larval stage onwards, spanning SOP specification, divisions and OSN differentiation (red bar). The late *peb-GAL4* driver is expressed only after OSNs begin to differentiate (green bar). **c** Examples of the six phenotypic categories. Left: lateral view of the fly head, with the eyes and antennae marked by arrows. When *Eip93F* RNAi is induced with the constitutive driver, both antennae and eyes display morphological defects, in addition to loss of eye pigmentation. Right: right antennal lobes of control and the indicated RNAi animals expressing an *Or67d*-CD8a:GFP reporter (green), which labels the axons of the Or67d and Or82a OSNs that innervate the DA1 and VA6 glomeruli (yellow arrows), respectively. RNAi phenotypes of specific genes with the constitutive driver include global glomerular morphology defects, axon density alteration, imprecise axon targeting, precise targeting errors and/or other defects (indicated by the white arrows). Scale bar = 20 μm. **d** Phenotypic classification of the 35 high confidence screen hits when the RNAi transgenes were driven by either the constitutive or late driver. Each gene is classified under six phenotypic categories in three levels of phenotypic severity

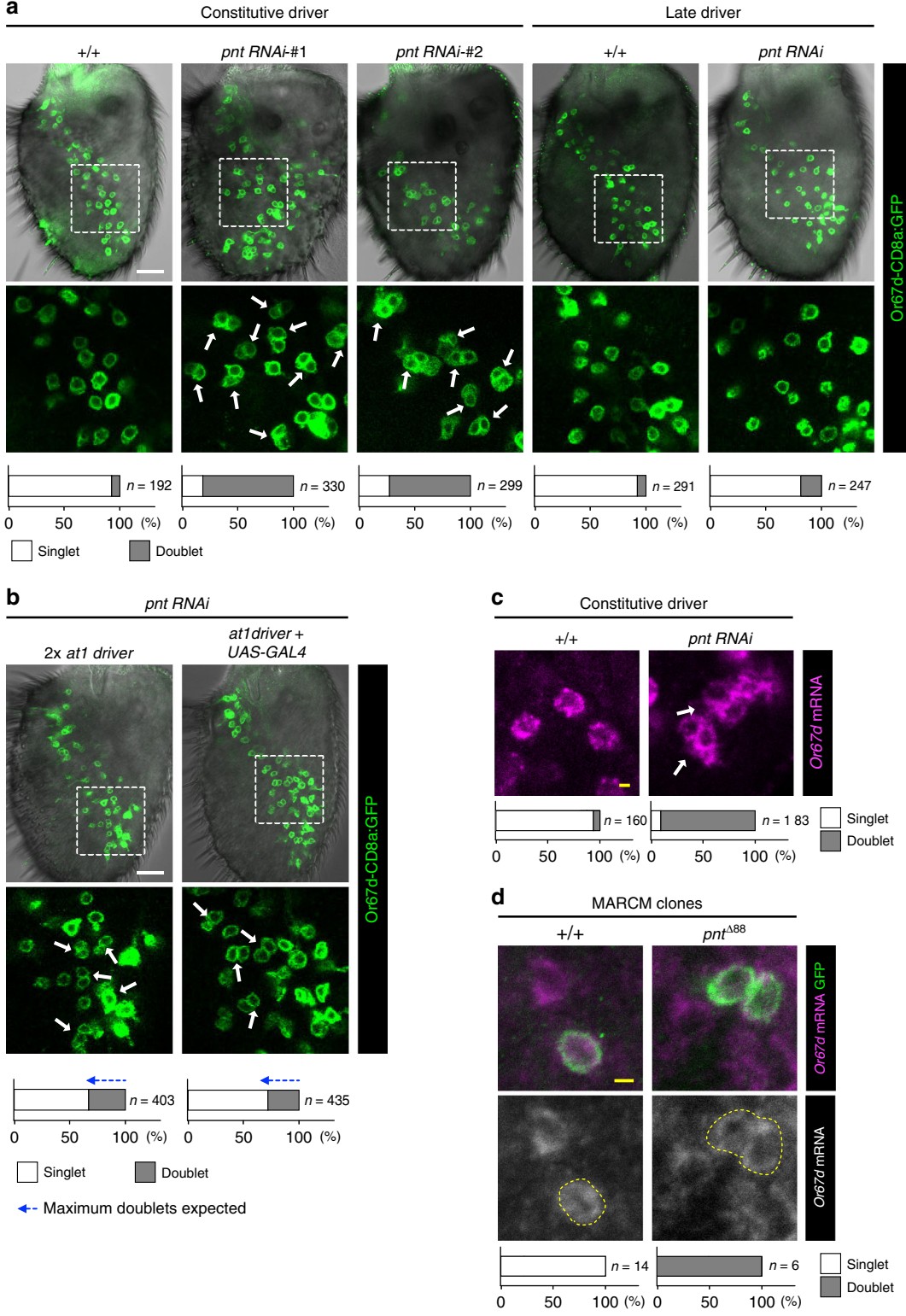

upregulated in the at1 SOPs just prior to their division (Fig. 8a). Its expression is immediately turned off after cell division—as the daughters of this SOP (pIIa and pIIb) are devoid of Pnt immunoreactivity—and resumes in all progeny of pIIa that form support cells. In the neural sublineage derived from pIIb, Pnt is only expressed in pNa and both of its daughters (Naa and Nab). However, its expression is rapidly downregulated in Nab cells, leaving Naa (marked by Svp) the only terminal cell in the neural sublineage to express this transcription factor (Fig. 8a, b).

**Pnt is essential to distinguish Naa and Nab neuron fates**. To understand the genesis of the duplicated Or67d OSNs when Pnt is lost, we generated GFP-labeled wild-type or *pnt* RNAi cell clones and examined the expression of diagnostic cell markers at the eight-cell stage. Although we were unable to selectively label the at1 lineage in these experiments, we could identify putative eight-cell at1 clones by their spatial distribution in the pupal antenna (Fig. 5a) and their characteristic expression of Sens, as determined by our lineage mapping (Fig. 5b, c). Comparison of

**Fig. 7** Loss of *pnt* leads to duplication of Or67d OSNs. **a** Phenotypes of control and *pnt* RNAi when driven with the constitutive and late drivers in antennae expressing the *Or67d*-CD8a:GFP reporter (green). Two independent *pnt* RNAi lines—KK100473 (#1) and JF02227 (#2), which target different genomic sequences within an exon common to all four *pnt* isoforms—result in doublets of GFP-positive neurons, compared to the predominant singlets of these neurons in controls. Late induction of RNAi produces few doublets. The lower images show the magnified views of the region within the dashed box in the upper images. Quantifications of the fraction of singlets/doublets are shown in the stacked histograms at the bottom. White scale bar = 20 μm in this and other panels. **b** *pnt* RNAi when driven with at1 driver in antennae expressing the *Or67d*-CD8a:GFP reporter produces OSN doublets. Due to the relatively weak late expression of the at1 driver, we increased GAL4 dosage either through inclusion of two copies of the driver (left) or an auto-amplifying *UAS-GAL4* transgene (right). While the driver is restricted to about half of the at1 lineages (Fig. 4b), *Or67d*-CD8a:GFP is expressed in both at1 and, ectopically, ab5 [13]. Based upon the 2:1 ratio of the numbers of at1 and ab5 sensilla[37], the maximum percentage of doublets expected in these RNAi experiments is ~33%, shown by the length of the blue arrows with dashed lines (i.e. half of the at1/Or67d neurons, which represent 66% of all the neurons labeled by this reporter); this is the approximate percentage observed in both genetic manipulations. **c** Control at1 sensilla have a single *Or67d* mRNA-expressing OSN (magenta). In *pnt* RNAi antennae, *Or67d* mRNA is detected in doublets of OSNs. Yellow scale bar = 2 μm in this and other panels. **d** Representative MARCM clones of control and *pnt* null mutant neurons, marked with CD8a:GFP (green). *Or67d* mRNA-expressing OSNs (magenta) in control clones are singlets, while those in *pnt* mutant clones are doublets

eight-cell wild-type and *pnt* RNAi clones revealed indistinguishable patterns of Sens and Ham expression in 4 and 2 cells, respectively (Fig. 8c and Supplementary Fig. 5b). These observations suggest that loss of Pnt does not lead to conversion of Pnt-positive support cell lineages into neuronal fates, nor does it transform the Pnt-positive pNa lineage into the pNb lineage. By contrast, expression of Svp, which labels uniquely the Naa cell in controls, is lost in almost all (16/17) *pnt* RNAi clones examined (Fig. 8c). This observation suggests a model in which Pnt functions in the Naa cell to induce Svp expression, which distinguishes its fate from the Nab cell. In the absence of Pnt, the pNa cell divides to produce two Svp-negative Nab cells, which differentiate to form the observed doublet of Or67d OSNs.

We tested this model first by asking whether Pnt was sufficient to promote Naa fate by misexpressing Pnt in SOP clones (Fig. 8d). Many of the intermediate daughter cells within each SOP lineage expressed Svp, consistent with the ability of Pnt to induce this marker of Naa cells. We were unable, however, to examine later phenotypes as the ectopic expression of Pnt and/or Svp blocked lineage development at the five-cell stage (Fig. 8d). Next, we examined the role of Svp itself, which has only previously been used as a marker of Naa fate. Notably, *svp* RNAi in the antenna phenocopies *pnt* RNAi, giving rise to doublets of Or67d OSNs (Fig. 8e). Although misexpression of *svp* throughout the SOP lineage did not lead to a significant reduction in Or67d OSN number, it could partially suppress the formation of Or67d OSN doublets in a *pnt* RNAi background (Fig. 8f). These results indicate that Svp functions downstream of Pnt to define Naa fate, but is insufficient alone to confer Naa fate when ectopically expressed in other cell types.

**MAPK-independent function of Pnt in Naa/Nab differentiation.** Pnt belongs to the ETS transcription factor family[45,46]. Studies of both *Drosophila* and vertebrate homologs have demonstrated that these transcription factors are MAP kinase-regulated nuclear effectors of the EGFR, JNK and p38 signaling pathways in diverse developmental contexts[47,48]. To determine whether Pnt is regulated in a similar manner in the at1 lineage, we performed RNAi (or used putative dominant-negative versions) for several ligands, receptors and cytoplasmic components of each of these signal transduction cascades. Although the size of the antenna and number of OSNs was altered in some cases, we never observed doublets of Or67d OSNs beyond the low frequency observed in controls (Supplementary Fig. 5c). We extended this analysis by simultaneous interference of components from two of these signaling cascades, but again failed to observe a *pnt*-like phenotype in Or67d OSN specification (Supplementary Fig. 5c). Although we cannot exclude that these negative results reflect insufficient RNAi for some genes, our

observation that none of the 17 MAPK signaling components tested gave a *pnt*-like phenotype suggests that upstream regulation of Pnt in the at1 SOP lineage occurs in a different way to its other known developmental roles.

**A universal requirement for Pnt in OSN lineages.** The at1 lineage is unique because it is the only sensillar subtype that harbors a single OSN (of Nab fate). All other olfactory sensilla contain two (Nab/Nba), three (Naa/Nab/Nba) or a maximum of four (Naa/Nab/Nba/Nbb) OSNs (Fig. 9a). The at1 Naa cell that adopts Nab fate in the absence of Pnt is, in wild-type flies, originally destined for apoptosis (Fig. 9b). To test if Pnt has a broader function in SOP lineages, we examined the expression of receptors in other sensillar classes in control and *pnt* RNAi antennae by RNA in situ hybridization. In the four classes of two-OSN sensilla tested (ab2, ab3, ai2, ac3), duplication of the Nab-derived OSN was observed, while expression of receptors in the other (Nba) OSN was unchanged (Fig. 9c, d and Supplementary Fig. 6a, b). A similar phenotype occurs in the four-OSN ab1 sensilla: here, the Naa cell appears to transform from one OSN fate (*Or10a*-expressing) to another (*Or92a*-expressing), while the other two OSNs (Nba/*Or42b* and Nbb/*Gr21a*) are unaffected (Fig. 9e and Supplementary Fig. 6a–c). These results are consistent with a conserved role of Pnt in the SOP lineages giving rise to these different sensillar classes.

To expand this analysis to all antennal olfactory SOPs, we compared olfactory receptor gene expression levels in control and *pnt* RNAi antennae by RNA-sequencing. Loss of Pnt leads to an overall reduction in the number of antennal OSNs (Supplementary Fig. 6d)—with a commensurate decrease in transcripts for most olfactory receptors (Supplementary Data 1)—suggesting that it has an earlier, general function in SOP specification in the antennal disc. Despite this global effect, we reasoned that cell fate transformations within individual SOP lineages would be recognizable as changes in the relative expression of receptor genes of OSNs housed in the same sensillum (Fig. 9f and Supplementary Fig. 6e).

Indeed, by normalizing the transcript read counts of the receptors to the theoretical ratios of their underlying OSNs (see Methods and Supplementary Fig. 6e), we observed that all (12/12) two-OSN sensillar classes had augmented expression of the receptor expressed in Nab relative to the receptor in Nba (Fig. 9f). The change is most simply explained by a duplication of the Nab OSN, as already determined in a subset of sensilla (Fig. 9b, e, and Supplementary Fig. 6a). Similarly, in most (4/5) three-OSN sensillar classes and the four-OSN ab1 class, the increase of the Nab olfactory receptor expression relative to the Naa olfactory receptor expression is consistent with an Naa to Nab fate transformation (Fig. 9f). Only two exceptions were observed: in

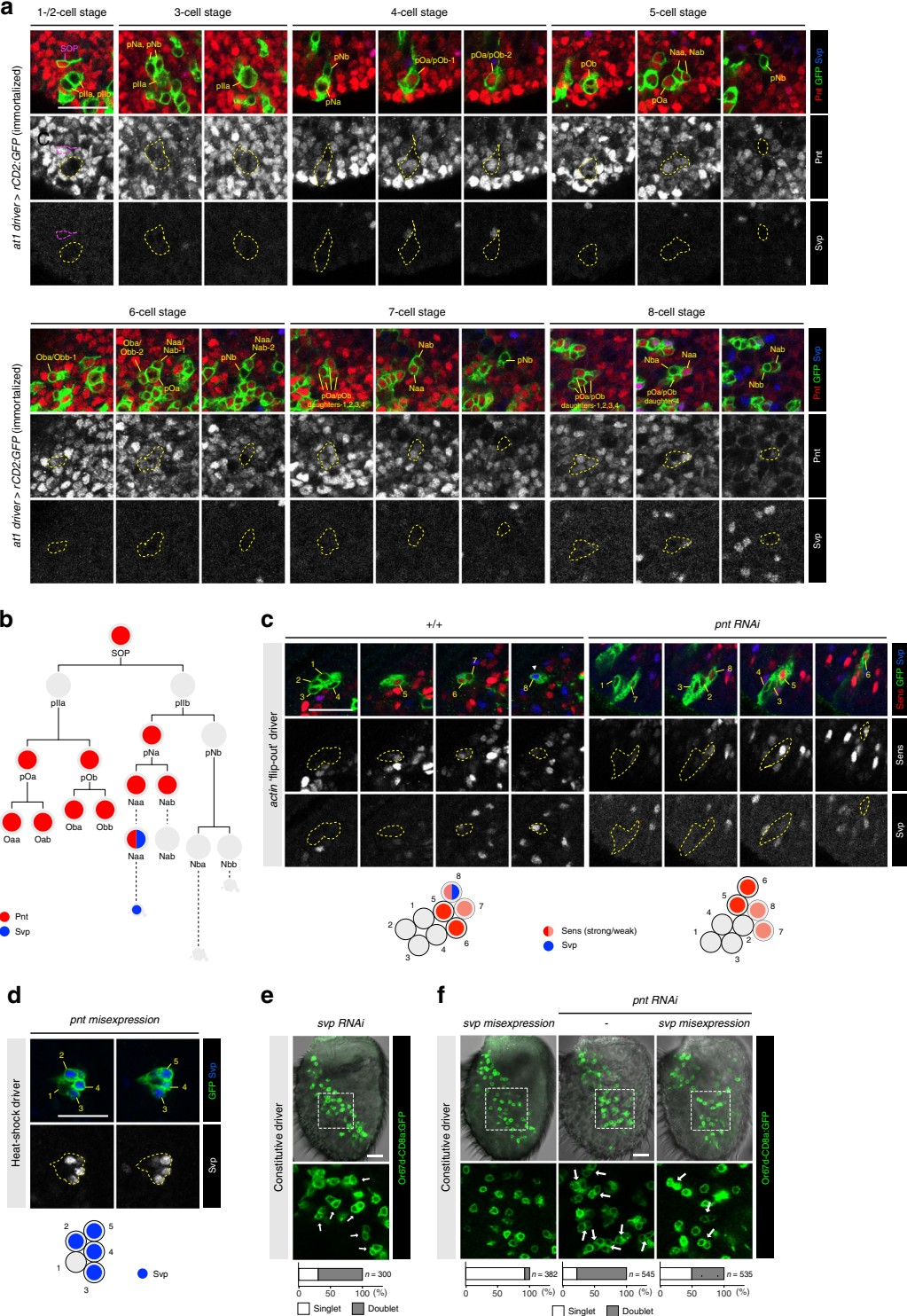

**Fig. 8** Pnt is dynamically expressed in the at1 lineage and required for Svp expression. **a** Expression profiles of Pnt (red) and Svp (blue) in at1 lineages visualized with the immortalized at1 driver (green). Scale bar = 20 μm in this and other panels. **b** Schematic summarizing the expression of Pnt and Svp in the at1 lineage. **c** Left: representative control 8-cell heat-shock clone (green) that shows the typical expression profile of the at1 lineage: two cells with stronger Sens expression (red) and two cells with weaker Sens expression, one of which co-expresses Svp (blue) (see Fig. 5b, c). Right: in a representative eight-cell *pnt* RNAi clone (green), Sens expression (red) is unaffected but Svp expression (blue) is absent. Schematics summarizing the expression of the markers in these clones are shown below the images. **d** Misexpression of Pnt in a five-cell clone (green) induces ectopic Svp expression (blue) in four of the cells. (Clones do not develop beyond this stage.) **e** *svp* RNAi antennae expressing the *Or67d*-CD8a:GFP reporter exhibit a large fraction of doublets of GFP-positive neurons (green). **f** Misexpression of *svp* alone (left) does not affect the normal specification of *Or67d*-CD8a:GFP-labeled OSNs. Two independently constructed *UAS-svp* transgenes were used in this experiment: *UAS-svp.II* (shown in this panel) and *UAS-svp1*. Misexpression of *svp* in the *pnt* RNAi background reduces the fraction of OSN doublets (middle and right)

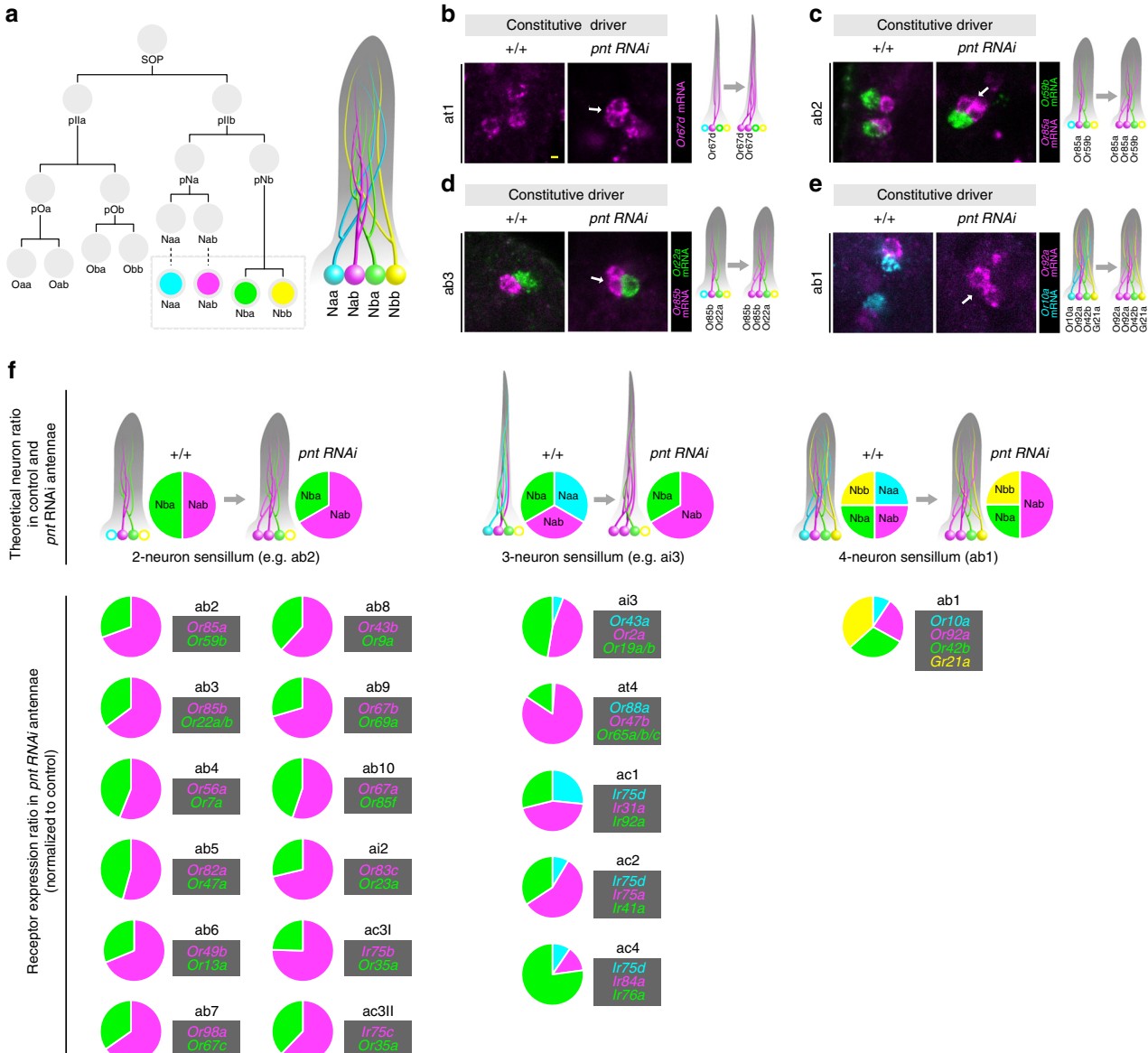

**Fig. 9** Loss of Pnt results in Naa-to-Nab transformations in diverse sensillar subtypes. **a** A sensillum can contain up to four OSNs through differentiation of Naa (cyan), Nab (magenta), Nba (green), Nbb (yellow) terminal daughter cells originating from a single SOP lineage. **b** Representative images of RNA FISH for *Or67d* (magenta) in at1 sensilla in control and *pnt* RNAi antennae. In *pnt* RNAi antennae, Or67d-expressing OSNs are duplicated (arrow). A schematic of the proposed Naa-to-Nab fate transformation is shown on the right (color scheme as in (**a**)). Scale bar = 2 μm. The open circles in this and other schematics represent OSN precursors that have undergone apoptosis. **c** Representative images of RNA FISH for *Or85a* (magenta) and *Or59b* (green) in ab2 sensilla in control and *pnt* RNAi antennae. In *pnt* RNAi antennae, Or85a OSNs (Nab) are duplicated (arrow), while Or59b OSNs (Nba) are unaffected. **d** Representative images of RNA FISH for *Or85b* (magenta) and *Or22a* (green) in ab3 sensilla in control and *pnt* RNAi antennae. In *pnt* RNAi antennae, Or85b OSNs (Nab) are duplicated (arrow), while Or22a OSNs (Nba) are unaffected. **e** Representative images of RNA FISH for *Or92a* (magenta) and *Or10a* (cyan) in ab1 sensilla in control and *pnt* RNAi antennae. In *pnt* RNAi antennae, Or92a OSNs (Nab) are duplicated (arrow), while Or10a OSNs (Naa) are lost. **f** Top: theoretical ratios of OSN types in 2-, 3- and 4-neuron sensilla in control and *pnt* RNAi antennae, assuming Naa-to-Nab fate transformation (i.e. loss of Naa OSNs, and duplication of Nab OSNs). Bottom: experimentally determined OSN ratios in all sensilla in *pnt* RNAi antennae using as a proxy the normalized ratios of olfactory receptor mRNA expression from antennal transcriptomes (see Supplementary Fig. 6e). In ab10, *Or49a* is reported to be coexpressed with *Or85f*[13], but transcript levels for this gene were below the cut-off applied during the analysis of these RNA-seq datasets

ac4 sensilla, *Ir84a* (Nab) expression is greatly diminished, potentially reflecting a role for Pnt in activation of this receptor gene; ab1 sensilla appear to have increased transcript levels for *Or42b* (Nba) and *Gr21a* (Nbb) relative to *Or92a* (Nab), highlighting a more complex function of Pnt in this unique four-OSN sensillar class. Globally, however, these transcriptomic data are consistent with a universal role for Pnt in distinguishing Naa from Nab in olfactory sensory lineages.

## Discussion

By combining genetic drivers labeling small subsets of precursor cells with methods for immortalization of expression patterns within defined temporal windows, we have generated a fate map of the complete peripheral *Drosophila* olfactory system. This resource adds a novel developmental perspective on the *Drosophila* olfactory circuitry, complementing the maps of glomerular innervations[13–15] and PN projections to higher brain centers[49].

While the concentric spatial organization of SOPs is partially maintained in the distribution of olfactory sensilla, there is only a limited relationship with the organization of the axon projections of OSNs in the antennal lobe. Future immortalization of additional *enhancer*-GAL4 drivers with restricted antennal disc expression should help improve the resolution of the fate map, and identify unique sensilla lineage markers in addition to the at1 driver. This information will be important to investigate the developmental mechanisms that act within a particular arc to specify up to 5–6 different SOP types.

Previous screening for transcriptional determinants of OSN fates identified a small set of factors that act in a combinatorial manner to activate or repress olfactory receptor expression in specific OSN classes[50–52]. Such factors are likely to act only at the end of more elaborate gene regulatory networks that ensure the specification of SOP type, and determination and coordination of OSN receptor expression and axon targeting[53,54]. Our genome-wide, constitutive RNAi screen of transcriptional regulators has identified a large number of new molecules that are likely to function in several of these processes. We focused on the role of the ETS homolog, Pnt, because of its unique mutant phenotype, which reveals a role in limiting, rather than determining, Or67d neuron specification. With our at1 lineage marker, protein expression and gain- and loss-of-function analyses, we provide evidence that this transcription factor has a switch-like function in distinguishing the terminal Svp-expressing Naa cell from its Svp-negative sibling Nab. Interestingly, this role of Pnt appears to be distinct from other functions of this transcription factor where it serves as a nuclear read-out of various MAPK signaling pathways. Our antennal transcriptomic analysis indicates that this role of Pointed is likely to be universal in olfactory sensilla. Moreover, the switch-like function is not the only role of Pnt in the antenna, as it also contributes to the specification of the correct global number of SOPs, and may more directly regulate the expression of specific olfactory receptor genes (e.g. *Ir84a*). Pnt's broad expression in the non-neuronal sublineage suggests it could also participate in support cell development.

With our OSN lineage driver, we may now exploit single-cell RNA-seq and chromatin profiling technologies to examine the gene expression and epigenetic states of the at1 lineage from birth to maturity, and how these may be influenced by internal state and environmental conditions[55]. While cellular-resolution level transcriptomic/epigenomic data are undeniably important to understand neural development, the combination of these with methods for visualizing specific lineages in vivo is essential for a complete view of how structural and functional diversity develops in the nervous system.

## Methods

**Drosophila stocks**. *Drosophila* stocks were maintained on standard cornmeal medium under a 12:12 h light/dark cycle at 25 °C, unless otherwise stated. The mutant and transgenic lines used are listed in Supplementary Table 2. Additional GAL4 drivers screened with the immortalization system to identify OSN lineage markers (but subsequently discarded due to irreproducible labeling) are listed in Supplementary Table 5. Targets for the RNAi screen were collated from (i) transcription factors (FlyTF.org[56]), (ii) proteins involved in chromatin-related processes (FlyTF.org[56]), and (iii) embryonic patterning genes (gap genes, pair-rule genes and segment polarity genes listed in The Interactive Fly (sdbonline.org/sites/fly/aimain/5zygotic.htm)).

**GAL4 driver immortalization**. Embryos of flies arising from the desired genotypic cross were collected over 8 h and allowed to develop at 19 °C until the larvae began pupation (typically after ~7 days under our culture conditions). All non-white pupae (representing older ages) were discarded, while the other larvae and white pupae (i.e. aged ~0–8 h BPF) were subjected to 24 h at 29 °C to heat-inactivate the GAL80ts. Animals were returned to 19 °C to continue development until eclosion after a further ~6–7 days. For late immortalization of *en-GAL4*, heat-inactivation was carried out on pupae aged ~9–15 h APF.

For at1 lineage analysis in the adult, embryos were collected over a period of either 9 or 15 h (representing overday and overnight collections, respectively) at 19 °C, and aged until the larvae began pupation. All the older pupae were discarded such that the remaining larvae/pupae (aged ~0–9 h BPF or ~0–15 h BPF for the different collections) were placed at 29 °C to heat-inactivate the GAL80ts (the stability of at1 driver expression allows the application of a wider age window than for other GAL4 drivers). The pupae were returned to 19 °C to continue development until they were ready to be assayed.

The analysis of at1 lineage development in the pupae was performed as above, but to ensure tight synchronization of development, only white pupae (aged ~0 h APF) were collected. These were subjected up to a maximum of 24 h heat-inactivation of GAL80ts at 29 °C, before they were returned to 19 °C for development. Pupal antennae were dissected at specific time points between 18 h APF and 27 h APF (i.e. within or after the heat-inactivation window).

Each experiment involved the use of 5–20 animals, repeated 1–3 times to ensure the consistency of the results. For at1 lineage tracing experiments, 10–30 lineages were examined per time point for each antibody combination.

**RNA interference**. To maximize the efficiency of RNAi, larvae of the desired genotype were raised at 27 °C (which enhances GAL4 induction) from 24 h after egg-laying until the animals were ready to be analyzed. For the minority of transgenic RNAi lines where this treatment resulted in pupal lethality, larvae were allowed to develop at 25 °C.

**Clone generation**. Heat-shock clones and MARCM clones were generated according to established methods[44,57]. Embryos were collected over 15 h and raised at 25 °C until 48–63 h after larval hatching. These larvae were subjected to 30 min heat shock in a 37 °C water bath and returned to 25 °C to continue development. For the examination of antennal SOP lineage development, white pupae were collected to re-synchronize their developmental timing and aged until 20–24 h APF. For each experimental condition, 20–50 clones from at least 10 antennae were examined.

**Histology and immunocytochemistry**. Immunofluorescence on adult brains was performed according to a standard protocol[58]. Immunofluorescence and RNA fluorescence in situ hybridization (FISH) on whole-mount adult antennae were performed using standard protocols[59] with slight modifications: (i) antennae were fixed for 3 h at 4 °C, (ii) RNA probes were denatured at 80 °C for 10 min and (iii) incubation with anti-Digoxigenin (DIG)-POD or anti-Fluorescein (FITC)-POD was for 36 h.

For immunofluorescence on larval antennal discs and pupal antennae, animals were placed in phosphate-buffered saline (PBS) and severed below the head to expose the discs or antennae before fixation in 4% paraformaldehyde in PBS + 0.2% Triton X-100 (PBT) for 1.5 h at 4 °C. Following ~5–10 quick rinses with PBT, excess tissues were removed. Samples were blocked in 5% normal goat serum diluted in PBT for 1 h at room temperature (RT) before incubation in primary antibodies (diluted in blocking solution) for 48 h at 4 °C. After five rounds of washes with PBT over 3 h, secondary antibodies (diluted in blocking solution) were added and left for 48 h at 4 °C. Following another five rounds of washes with PBT over 3 h, samples were equilibrated overnight at 4 °C in Vectashield mounting medium. During mounting onto a bridged slide, all unwanted tissues and head structures were cleaned from the antennal discs and pupal antennae. The antibodies used in this work are listed in Supplementary Table 3. All microscopy was performed using a Zeiss LSM 710 laser scanning confocal microscope. The raw confocal images were processed (cropping, brightness/contrast adjustment, and color channel separation) in Fiji[60].

For quantification of OSNs (based on immunofluorescence and FISH signals), 10–30 antennae were scored.

**Molecular biology**. The construction of plasmid templates and synthesis of DIG- and FITC-labeled RNA probes were performed using standard protocols[59,61]. The primers used to amplify the desired cDNA or genomic fragments (or previously described probes) are listed in Supplementary Table 4.

**Total RNA extraction from antennae**. Antennal RNA was extracted from four biological replicates of *pnt* RNAi flies (constitutive driver crossed to *UAS-pnt*RNAi-KK100473) and their paired controls (constitutive driver crossed to the same strain without the RNAi transgene: VDRC control 60100). For each pair of biological replicates, animals were grown under identical conditions and RNA was extracted in parallel.

For each biological replicate, the third antennal segments from 500 to 1000 flies were harvested via snap-freezing in a mini-sieve[62]. Antennae were transferred to a 1.5 ml Eppendorf tube on ice and homogenized manually with a tissue grinder. Total RNA was extracted from the homogenized antennae using a standard TRIzol/chloroform protocol. Briefly, TRIzol reagent was added to a final volume of 1 ml and the tube shaken vigorously manually for 1 min. After chilling on ice for 5 min and centrifugation (13,000 rpm for 15 min at 4 °C, for all subsequent centrifugations) the supernatant was transferred to a new tube. Two hundred microliters of chloroform was added, and the tube shaken vigorously for 15 s and left at RT for 3 min, before re-centrifugation. The upper aqueous phase was

transferred to a new tube, and RNA was precipitated with an equal volume of prechilled isopropanol. The mixture was incubated for 10 min at RT and centrifuged for 15 min. The resultant RNA pellet was washed with 1 ml prechilled 75% ethanol, air-dried, and reconstituted in 20 μl water.

**RNA library preparation and sequencing**. RNA quality was assessed on a Fragment Analyzer (Advanced Analytical Technologies, Inc.); all RNAs had an RQN of 7.9–10. From 100 ng total RNA, mRNA was isolated with the NEBNext Poly(A) mRNA Magnetic Isolation Module. RNA-seq libraries were prepared from the mRNA using the NEBNext Ultra II Directional RNA Library Prep Kit for Illumina (New England Biolabs). Cluster generation was performed with the resulting libraries using the Illumina TruSeq SR Cluster Kit v4 reagents and sequenced on the Illumina HiSeq 2500 using TruSeq SBS Kit v4 reagents (Illumina). Sequencing data were demultiplexed using the bcl2fastq Conversion Software (version 2.20, Illumina).

**RNA-seq data analysis pipeline**. Purity-filtered reads were adapters- and quality-trimmed with Cutadapt (version 1.8 [63]). Reads matching to ribosomal RNA sequences were removed with fastq_screen (version 0.11.1). Remaining reads were further filtered for low complexity with reaper (version 15-065 [64]). Reads were aligned to the *Drosophila melanogaster* BDGP6.86 transcriptome using STAR (version 2.5.3a[65]) and the estimation of isoform and gene abundance was computed using RSEM (version 1.2.31 [66]). Gene-level estimated counts from RSEM were used as input for the statistical analysis. Gene-level TPM (transcripts per kilobase million) from RSEM were used for the calculations of read count ratios between genes expressed in a common sensillar class.

Statistical analysis was carried out using R (version 3.4.4). The *pnt* RNAi antennae were compared to controls with a paired samples design. The RSEM gene-level data were read in with the Bioconductor package tximport (version 1.6.0.[68]). Fold changes and *p* values were calculated with the R Bioconductor package DESeq2 (version 1.18.1), which uses negative binomial GLM fitting and Wald statistics[68]. Genes with zero estimated counts in all samples were removed prior to normalization and model fitting, leaving 14,233 genes in the analysis. Default settings were used for the estimateSizeFactors() and estimateDispersions() functions. For multiple testing correction, the *p* values were adjusted together by the Benjamini−Hochberg method, which controls the false discovery rate[69]. The independent filtering option for *p* value adjustment was turned off.

**Reporting summary**. Further information on experimental design is available in the Nature Research Reporting Summary linked to this article.

## Data availability
RNA-seq data are available in GEO (Accession GSE113997). All other data supporting the findings of this study are available from the corresponding author on request.

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

## Acknowledgements

We are grateful to Mattias Alenius, Thomas Auer, Konrad Basler, Hugo Bellen, Yu Cai, Marco Cantoni, Ya-Hui Chou, Christian Fankhauser, Veit Grabe, Fisun Hamaratoglu, Yasushi Hiromi, Michael Hoch, Yuh-Nung Jan, Andrew Jarman, Kaan Mika, Marco Milán, Pavan Ramdya, Silke Sachse, Kuo-Ting Tsai, Pelin Volkan, Leslie Vosshall, Ryohei Yagi, Petra zur Lage, the Bloomington *Drosophila* Stock Center (NIH P40OD018537), the Vienna *Drosophila* Resource Center, the KYOTO Stock Center (Kyoto DGGR of Kyoto Institute of Technology), the Developmental Studies Hybridoma Bank (NICHD of the NIH, University of Iowa), and the Lausanne Genomic Technologies Facility for their generous sharing of reagents and resources. We thank Scott Barish, Sebastian Cachero, Yu Cai, Erika Dona, Pelin Volkan and members of the Benton laboratory for discussions and comments on the manuscript. Research in R.B.'s laboratory is supported by the University of Lausanne, an ERC Consolidator Grant (615094) and the Swiss National Science Foundation (31003A_166646).

## Author contributions

P.C.C. conceived the project, designed, performed, and analyzed most experiments. S.C. performed RNA in situ hybridization and prepared samples for RNA-seq. L.W. performed statistical analyses of RNA-seq data. R.B. supervised the project. P.C.C. and R.B. wrote the paper, with input from other authors.

## Additional information

**Competing interests:** The authors declare no competing interests.

