## [Peer Review File · Nature Communications]

Reviewers' comments:

Reviewer #1 (Remarks to the Author):

In this paper, Dr. Benton and colleagues used a genetic immortalization method to carefully map sensory neuron lineages, characterize the temporal patterns of cell divisions in some of these lineages. They next focused on at1 lineage and conducted a RNAi screen to search for molecules that may regulate OSN fate specifications. In the end they identified and verified that Pointed may function as a universal switch of OSN class specification. This paper can be divided into two parts. In the first part (Fig 1-Fig5, Supplementary Fig 1-3), they carefully mapped the SOP of different OSN classes, and then tried to map the distribution of SOP in the arch identity of antennal discs to the sensilla distribution in the antennae and OSN targeting in the AL. This work will update, revise and expand our current knowledge of the fate determination of SOPs in the antennal disc and bridge the SOP map in the disc to OSN map in the antenna. The second part (Fig 6-Fig 8, Supplementary Fig. 4-6) was first focusing on the molecular control of at1 lineage that Pointed involved. They then demonstrated a universal role of Point on switching the fates of OSNs in the same sensillum. This paper is composed of tremendous work with high quality results. It will be a fundamental work for future OSN studies. However I did have some concerns as listed below.

(Major concerns)

1. (Fig 1) Would the immortalization labeling system label additional cells in pupal brains, which eventually innervate adult ALs? For instance, the labeled OSN axons in DA1 are very sparse (Fig 1, raw 4, adult AL/immortalized). Are they from some other cells innervating the AL or only a small portion of Or67d OSNs expressing en-GAL4? The labeled processes in the raw 4 of Fig 2d also point out such possibility. Some of these processes look like from other cells and unlike OSN axons.
2. I am glad that above concern was partly taken care of as shown in FigS2. However, some processes still can be observed in the glomeruli of GMR13B04 and GMR33G07 brains from the antennae-removed flies. Are they from glia or some neurons? Would they complicate the results in Fig 2d and thus Fig 2e and 2f?
3. (Fig 2) The antero-posterior indicators of antennal disc shown in Fig 2a, 2c and 2f are incorrect. Actually, the antennal disc and the eye disc have opposite A-P axis. This is obvious that en-GAL4-positive cells, thus should be in the posterior compartment of the antennal disc, locate in the "upper half" of the antennal disc in Fig 1d.
4. (Fig 2) The number of Amos- or Ato-positive cells in each arc of 2h APF antennal disc look similar (Fig 2b). But the number of OSNs derived from each arc looks very different (Fig 2f). I was wondering if authors have any idea about this.
5. (p. 26, line 1173) In the figure legend of Fig 3, it was mentioned as ".....indicates mono-/polyglomerular OSNs,.....". Did the authors mean OSNs mapped to "polyglomerular PNs"? If so, how were mono-glomerular OSNs mapped to polyglomerular PNs?
6. (p. 7, line 277-279) The authors mentioned that they didn't observe obvious clustering of glomeruli innervation by OSNs derived from the same arc (Fig 3c). However the glomeruli targeted by OSNs derived from the same look like loosely clustered. I was wondering if the author could estimate the relative distance between glomeruli innervated by OSNs derived from the same arc and between these glomeruli and a set of random picked glomeruli to see whether there are differences.
7. (p7, line 286-290) The authors proposed an interesting model to explain their observation on OSN and adPN lineages. However, I am confused how they determine which OSNs and PNs belong

to ancestral olfactory circuit. Probably they could offer some references or have a statement as a supplementary note.

8. The major difference on lineage/cell fate determination between this work and Endo et al (2012, Nat Neurosci.) is that Naa and Nab undergo cell death in at1 SOP that produces Or67d OSN. Because at1 is the only sensillum with one OSN, I do not see obvious conflict between current work and Endo et al (2012, Nat. Neurosci.). However, from 4-cell to 8-cell stage, pOb, Oab, and Obb did not express Pon in Endo et al (2012, Nat Neurosci.) but this work found that these cells are Pon-positive (Fig S3-2) and marked them as Pon-positive in Fig 5c. I am not sure whether support cells also have such big difference between at1 SOP and other SOPs.

9. (Fig 8 and FigS6) The antennal transcriptomic analysis was used to determine the fate switch among OSNs in a given sensillum. It is probably the most efficient way to address the general role of Pnt. However, the result cannot distinguish whether the specific Nab-derived OSN remained one cell but has much higher OR expressions or the number of this OSN became two in pnt RNAi antennae.

10. (p. 12, line 547-550) The authors made a strong claim that "birthplace does not confer OSNs with axon guidance properties that reflect the topology of their origins". This is an over simplified rejection to all possible mechanisms.

(1) Concentric arc organization is one of a few features to compare the peripheral map and the central OSN axon targeting map. In addition, it is not clear whether all SOPs of the same type of sesilla locate in the same arch and in a fashion like at1 SOPs.

(2) They didn't examine whether the peripheral and the central maps is associated by other features, for instance, anterior-posterior axis, dorsal-ventral axis, or combinations of any two possible axes.

(3) Some of the drivers shown in Fig 2e label Pb1, Pb2 and Pb3 sesilla. The SOPs that give birth of these OSNs locate in a small region of the antennal disc, which was not included in the arc organization shown in Fig 2 and Fig 3 but they indeed contributed to the spatial glomerular organization shown in Fig 3c. Therefore the conclusion on no correlation between the peripheral and the central maps is over simplified and not proper.

11. In the discussion (p. 12, line 551-553), they only emphasised the good side of immortalization method and avoided the potential bad side. This should be balanced by stating the caveats of this method, for instance, transience GAL4 expression in any cells around the AL during development. Especially when glia wrapping and penetrating the glomeruli remain in all AL examples shown in Fig S2. In addition, if the GAL4 expression is very low in a part of GAL4-positive cells, it may not drive the expression of UAS-FLP in these cells, thus won't immortalize the full pattern of original GAL4 expression during the heatshock time window.

12. There are overwhelming amount of data in current manuscript. So I don't expect the authors to do any additional experiments. However, also because too many results are included, current manuscript is not easy to be digested by general readers. The authors may consider to concise the manuscript by either sizing down the data or removing some data from main figures to supplementary figures.

(Minor points)

13. (Fig 5) The four cells from pIIa lineages express weak Sens at 7-cell and 8-cell stages (Fig 5b) but it was marked as no Sens in the scheme (Fig 5c).

14. A part of Fig. 6c figure legend reads like for Fig 6d. Fig 6d, 6e, and figure legend are for Fig 6e, 6f and 6g, respectively.

Reviewer #2 (Remarks to the Author):

This paper establishes a methodology based on the immortalization of dynamic drivers, which allows tracing different adult OSNs back to specific progenitor groups. They define various progenitor groups based on a combination of drivers. They find a driver to label specifically the at1 lineage. Then, the authors use this and other more general drivers to screen for molecular factors involved in OSN specification. This analysis found that Pointed is critical to promote the Naa fate in the at1 and other OSN lineages.

Overall, this paper delineates most OSN lineages and identifies a novel player for the binary sister fate decision mechanism. The logic flow is straightforward: it first establishes a sophisticated methodology and then employs it to study cell specification and to find a very clear phenotype. Analysis of this phenotype reveals the universal importance of Pointed as a fate regulator in the OSN specification.

Major concerns

1. The expression of the different drivers in Figure 2D is not clear, especially for those with a broader expression. Using a nuclear report seems critical to resolve the different patterns. Also, showing the split channels as supplementary data (at least for the drivers with broadest patterns) would also help.
2. The experimental conditions should always be stated both in the main text and figures. Sometimes it is very hard to follow which Gal4 driver was used in every experiment, especially in figure 8.
3. The use of statistics in this paper is definitely improvable: 1) the number of animals used for each experiment should be mentioned so that the reader can have an idea of the inter-individual variability for these phenotypes. 2) In those cases where the phenotype is just mild (Figure 6E), a statistical test is necessary to understand the differences. 3) The phenotypes in Figure 8 seem fully penetrant. Is that right? How many sensilla were analyzed?
4. The effect of Pnt seems not too obvious when the at1-driver is used to drive RNAi (Figure 6E). However, the authors use this driver to conclude that MAPK signal transduction is not involved in the Pnt phenotype. Is it possible that a similar phenotype could occur if this RNAi is driven by the more general drivers?

Minor points

1. Figure S1C is redundant.
2. Showing those 19 drivers that gave inconsistent expression as supplementary material could be useful in future experiments.
3. When discussing a combination of drivers can define different territories, it should be remarked that this also includes amos and ato.
4. Is the expression of the GMR82D08 driver the same in males and females? This enhancer is endogenously located in the X chromosome and could be affected by sexual dimorphism, especially considering that Or67d is involved in courtship (Kurtovic et al., 2007).
5. In the paragraph beginning in line 356, the specific drivers used for Gal4 expression should be named and discussed.
6. In figure 6d, the analysis with the late Gal4 driver should also show both RNAi.
7. There is a typo in line 508: "expression is consistent with an Naa..."

NCOMMS-18-27518-T: RESPONSE TO REVIEWERS

We thank the reviewers for their careful reading and constructive criticisms of our manuscript. Below, we provide responses to each of the raised issues.

Reviewer #1

In this paper, Dr. Benton and colleagues used a genetic immortalization method to carefully map sensory neuron lineages, characterize the temporal patterns of cell divisions in some of these lineages. They next focused on at1 lineage and conducted a RNAi screen to search for molecules that may regulate OSN fate specifications. In the end they identified and verified that Pointed may function as a universal switch of OSN class specification. This paper can be divided into two parts. In the first part (Fig 1-Fig5, Supplementary Fig 1-3), they carefully mapped the SOP of different OSN classes, and then tried to map the distribution of SOP in the arch identity of antennal discs to the sensilla distribution in the antennae and OSN targeting in the AL. This work will update, revise and expand our current knowledge of the fate determination of SOPs in the antennal disc and bridge the SOP map in the disc to OSN map in the antenna. The second part (Fig 6-Fig 8, Supplementary Fig. 4-6) was first focusing on the molecular control of at1 lineage that Pointed involved. They then demonstrated a universal role of Point on switching the fates of OSNs in the same sensillum. This paper is composed of tremendous work with high quality results. It will be a fundamental work for future OSN studies. However I did have some concerns as listed below.

(Major concerns)

1. (Fig 1) Would the immortalization labeling system label additional cells in pupal brains, which eventually innervate adult ALs? For instance, the labeled OSN axons in DA1 are very sparse (Fig 1, row 4, adult AL/immortalized). Are they from some other cells innervating the AL or only a small portion of Or67d OSNs expressing *en-GAL4*? The labeled processes in the row 4 of Fig 2d also point out such possibility. Some of these processes look like from other cells and unlike OSN axons.

RESPONSE: It is true that the immortalization labeling system will not, in most cases, cover the entire population of a particular class of OSNs. As mentioned in the text, immortalization of the at1 driver led to labeling of ~35/60 Or67d OSNs. The lack of complete coverage is because most of the enhancer-GAL4s are not intrinsically linked to specific progenitor fates. Regarding the sparse labeling of DA1 resulting from the immortalization of *en-GAL4*, we suspect that this reflects the fact that the driver is restricted to the posterior domain of the antennal disc, while the at1 (DA1 OSN) progenitors are distributed in both posterior and anterior domains.

The processes from other cells typically do not interfere with our analysis because their signals are very weak and diffuse within glomeruli, a signal that is quite distinct from that of OSN axons (we appreciate that it may be less easy to judge this from the Z-projected confocal stacks presented in the figures). Our major concern with respect to incorrect glomerular assignment was potential confounding signal from projection neurons and local interneurons; we have ruled this possibility out, however, through the deafferentation experiments shown in Supplementary Fig. 2b.

2. I am glad that above concern was partly taken care of as shown in FigS2. However, some processes still can be observed in the glomeruli of GMR13B04 and GMR33G07 brains from the antennae-removed flies. Are they from glia or some neurons? Would they complicate the results in Fig 2d and thus Fig 2e and 2f?

RESPONSE: The glia-like processes observed in the antennal lobes of deafferentated flies are less pronounced in the non-treated animals. We speculate that this may be due to the non-specific and low-level activation of FLP during the long post-deafferentation period (14 days) prior to staining.

We acknowledge that non-specific activation of the immortalization system may complicate downstream analysis. For this reason, we presented the data for glomerular labelling in the form of heatmaps (Fig. 1e and Fig. 2e) in order to clearly distinguish strong and consistent signals resulting from temporally restricted GAL4 expression, from those that are weak and inconsistent. Our SOP mapping relies principally on the strong/consistent signals, and only incorporates additional, intermediate glomerular signals in certain cases.

3. (Fig 2) The antero-posterior indicators of antennal disc shown in Fig 2a, 2c and 2f are incorrect. Actually, the antennal disc and the eye disc have opposite A-P axis. This is obvious that en-GAL4-positive cells, thus should be in the posterior compartment of the antennal disc, locate in the "upper half" of the antennal disc in Fig 1d.

RESPONSE: We thank the reviewer for pointing out this inadvertent error. The A-P axis on the figures has been reversed and, to avoid confusion, we have also indicated in Fig. 2c that the antennal disc A-P axis is of opposite polarity to the body A-P axis.

4. (Fig 2) The number of Amos- or Ato-positive cells in each arc of 2h APF antennal disc look similar (Fig 2b). But the number of OSNs derived from each arc looks very different (Fig 2f). I was wondering if authors have any idea about this.

RESPONSE: As mentioned in the figure legend, the antennal disc in Fig. 2b is from a 4 h APF preparation. We chose a different timepoint for this panel because Ato-expression is very weak at 2 h APF, making it difficult to show its

intercalating pattern with Amos. At 4 h APF, the central region of the antennal disc protrudes with most of the Amos-positive cells located in the deeper layer (thus, they are hidden from view in the Z-section). Moreover, the number of progenitors in different arcs may not necessarily predict the number of sensilla; it is possible that there are progenitors that are not specified into any known olfactory sensilla. As one potential example, we failed to map any olfactory sensilla to arc 4 (although this could simply be a limitation of the set of enhancer-GAL4 lines tested in the current study).

5. (p. 26, line 1173) In the Fig. legend of Fig 3, it was mentioned as ".....indicates mono-/polyglomerular OSNs,.....". Did the authors mean OSNs mapped to "polyglomerular PNs"? If so, how were mono-glomerular OSNs mapped to polyglomerular PNs?

RESPONSE: Yes, we meant “polyglomerular PNs” and have corrected our text accordingly. Monoglomerular OSNs may map to both monoglomerular and polyglomerular PNs that innervate a particular glomerulus. For example, the axons of Ir31a OSNs (derived from arc3) innervate VL2p, which is innervated both by dendrites from VL2p-specific adPNs, as well as L3/L4 polyglomerular PNs (described in Yu et al., PLOS Biol 2010).

6. (p. 7, line 277-279) The authors mentioned that they didn't observe obvious clustering of glomeruli innervation by OSNs derived from the same arc (Fig 3c). However the glomeruli targeted by OSNs derived from the same look like loosely clustered. I was wondering if the author could estimate the relative distance between glomeruli innervated by OSNs derived from the same arc and between these glomeruli and a set of random picked glomeruli to see whether there are differences.

RESPONSE: We have followed the suggestion of the reviewer and found that there is significant, albeit mild, clustering of glomeruli innervated by OSNs derived from only arcs 2, 5, and 7. We have adjusted our statement in the text and added a violin plot to depict the pairwise glomerular distances associated with each arc in Fig. 3c.

7. (p7, line 286-290) The authors proposed an interesting model to explain their observation on OSN and adPN lineages. However, I am confused how they determine which OSNs and PNs belong to ancestral olfactory circuit. Probably they could offer some references or have a statement as a supplementary note.

RESPONSE: We hypothesize that the ancestry of OSNs reflects the ancestry of the receptors they express: the older olfactory circuits are likely to be those that express the most deeply conserved receptors (comprising several members of the IR family (e.g., IR64a (DP1m glomerulus), which is present across many insect orders; Croset et al., PLOS Genetics 2016). By contrast, ORs are a more recently evolved family of olfactory receptors, based upon phylogenetic criteria

(Croset et al., PLOS Genetics 2016). The only exceptions to the family-wide rule are IR75b (DL2d) and IR75c (DL2v), which are probably the most recently appearing IRs (Prieto-Godino et al., Neuron 2017); we found it of interest that these are expressed in OSNs derived from SOPs in the outermost arc, and synapse with PNs born at the end of the lineage. As we indicate in the manuscript, this is not an absolute relationship, and the model we derive is speculative (as is often the case when making evolutionary arguments). We now clarify a basis for the assumptions of ancestry of olfactory circuits with a brief addition to the text.

8. The major difference on lineage/cell fate determination between this work and Endo et al (2012, Nat Neurosci.) is that Naa and Nab undergo cell death in at1 SOP that produces Or67d OSN. Because at1 is the only sensillum with one OSN, I do not see obvious conflict between current work and Endo et al (2012, Nat. Neurosci.). However, from 4-cell to 8-cell stage, pOb, Oab, and Obb did not express Pon in Endo et al (2012, Nat Neurosci.) but this work found that these cells are Pon-positive (Fig S3-2) and marked them as Pon-positive in Fig 5c. I am not sure whether support cells also have such big difference between at1 SOP and other SOPs.

RESPONSE: Endo et al. performed a remarkable characterization of SOP lineage development, based on random (heat shock-induced) clone labeling. The key benefit of using the immortalization labeling system in lineage mapping is that it allows us to restrict the analysis to specific sensillar lineages that develop in a synchronized fashion. This gives us a better opportunity to observe the subtle changes in transcription factor expression during cell division without interference from other SOP lineages that may have different expression dynamics. This is especially important for the analysis of Pon because it is degraded rapidly upon segregation into one sibling cell. Moreover, using our at1 lineage marker, we were able to determine the birth/apoptosis order of all the daughter cells in this lineage. Whether this is universally conserved across lineages remains to be determined. We note that with our own random clones (Supplementary Fig. 3a-b), we observed some SOP lineages with transcription factor profiles that deviates from that of the at1 lineage. Therefore, we are fairly confident that there will be differences in other SOP lineages, whose precise nature will require other lineage-specific drivers.

9. (Fig 8 and FigS6) The antennal transcriptomic analysis was used to determine the fate switch among OSNs in a given sensillum. It is probably the most efficient way to address the general role of Pnt. However, the result cannot distinguish whether the specific Nab-derived OSN remained one cell but has much higher OR expressions or the number of this OSN became two in pnt RNAi antennae.

RESPONSE: We fully agree with the reviewer that our transcriptomic analysis is inadequate to address the possibility of increased OR expression in the cells due to *pnt* RNAi. Our intention was to use the transcriptomic data to supplement the

RNA FISH analysis (now Fig. 9b-e and Supplementary Fig. 5a-c) where Nab duplication is seen in *all* sensillar subtypes examined. In addition, our observation that the increase in OR expression in Nab cells is often accompanied by a decrease of OR expression from the corresponding Naa cell (e.g., ai3, at4, ac2, ab1) is consistent with the proposed Naa-to-Nab fate switch.

10. (p. 12, line 547-550) The authors made a strong claim that "birthplace does not confer OSNs with axon guidance properties that reflect the topology of their origins". This is an over simplified rejection to all possible mechanisms.

(1) Concentric arc organization is one of a few features to compare the peripheral map and the central OSN axon targeting map. In addition, it is not clear whether all SOPs of the same type of sesilla locate in the same arch and in a fashion like at1 SOPs.

(2) They didn't examine whether the peripheral and the central maps is associated by other features, for instance, anterior-posterior axis, dorsal-ventral axis, or combinations of any two possible axes.

(3) Some of the drivers shown in Fig 2e label Pb1, Pb2 and Pb3 sesilla. The SOPs that give birth of these OSNs locate in a small region of the antennal disc, which was not included in the arc organization shown in Fig 2 and Fig 3 but they indeed contributed to the spatial glomerular organization shown in Fig 3c.

Therefore the conclusion on no correlation between the peripheral and the central maps is over simplified and not proper.

RESPONSE: We have drastically cut our Discussion to conform to the 5000 word limit for the manuscript, resulting in the removal of this claim. We agree that our 'deleted' statement is an oversimplified rejection to all possible mechanisms and we thank the reviewer for the insights.

11. In the discussion (p. 12, line 551-553), they only emphasized the good side of immortalization method and avoided the potential bad side. This should be balanced by stating the caveats of this method, for instance, transience GAL4 expression in any cells around the AL during development. Especially when glia wrapping and penetrating the glomeruli remain in all AL examples shown in Fig S2. In addition, if the GAL4 expression is very low in a part of GAL4-positive cells, it may not drive the expression of UAS-FLP in these cells, thus won't immortalize the full pattern of original GAL4 expression during the heatshock time window.

RESPONSE: It is correct that the immortalization system is dependent on the strength and coverage of GAL4 expression within the heat-inactivation window. Our sole emphasis on the positive aspects of the immortalization method in the Discussion has now been removed in order to comply with word limits. However, we hope that within the context of the Results, our numerous control experiments, the judicious selection of only 6 out of 25 GAL4 lines with more consistent spatial/temporal expression patterns, and the use of heatmaps in our analysis (Fig. 1e and Fig. 2e) help to clearly illustrate the caveats of this system and the strategies to circumvent them.

12. There are overwhelming amount of data in current manuscript. So I don't expect the authors to do any additional experiments. However, also because too many results are included, current manuscript is not easy to be digested by general readers. The authors may consider to concise the manuscript by either sizing down the data or removing some data from main figures to supplementary figures.

RESPONSE: We thank the reviewer for acknowledging the breadth of our data. We did not spot an obvious way to remove data, or arbitrarily move some data panels to the supplementary information. To fulfill editorial requirements, we have cut the text drastically (principally in the Discussion), revised it for clarity, and, in response to Reviewer #2's comments, reorganized some of the figures and added new panels (Supplementary Fig. 2a and Supplementary Fig. 4).

(Minor points)

13. (Fig 5) The four cells from pIIa lineages express weak Sens at 7-cell and 8-cell stages (Fig 5b) but it was marked as no Sens in the scheme (Fig 5c).

RESPONSE: We thank the reviewer for pointing out this mistake and have updated the schematic in Fig. 5c.

14. A part of Fig. 6c Fig. legend reads like for Fig 6d. Fig 6d, 6e, and Fig. legend are for Fig 6e, 6f and 6g, respectively.

RESPONSE: We thank the reviewer for spotting the mix-up in the figure legend. In addition to the correction, we have now split Fig. 6 into two figures (new Fig. 6 and Fig. 7) to be able meet the 350-word limit for individual figure legends.

Reviewer #2

This paper establishes a methodology based on the immortalization of dynamic drivers, which allows tracing different adult OSNs back to specific progenitor groups. They define various progenitor groups based on a combination of drivers. They find a driver to label specifically the at1 lineage. Then, the authors use this and other more general drivers to screen for molecular factors involved in OSN specification. This analysis found that Pointed is critical to promote the Naa fate in the at1 and other OSN lineages.

Overall, this paper delineates most OSN lineages and identifies a novel player for the binary sister fate decision mechanism. The logic flow is straightforward: it first establishes a sophisticated methodology and then employs it to study cell specification and to find a very clear phenotype. Analysis of this phenotype reveals the universal importance of Pointed as a fate regulator in the OSN specification.

Major concerns

1. The expression of the different drivers in Fig. 2D is not clear, especially for those with a broader expression. Using a nuclear report seems critical to resolve the different patterns. Also, showing the split channels as supplementary data (at least for the drivers with broadest patterns) would also help.

RESPONSE: We chose a membrane marker because it encapsulates the nuclear signals of Amos and Dac (used to define the arcs) without overlapping with them. Based upon our analysis of raw confocal images, we are confident in our assignments, but we appreciate that the resolution of expression patterns may not be easy to distinguish in the very small panels presented to the reader. To ameliorate this issue, we have now included the GFP channel of those panels in Supplementary Fig. 2a along with the boundaries of the arcs.

2. The experimental conditions should always be stated both in the main text and figures. Sometimes it is very hard to follow which Gal4 driver was used in every experiment, especially in Fig. 8.

RESPONSE: We used the constitutive GAL4 driver for all of the RNAi experiments unless otherwise stated in the main text (i.e., late GAL4 driver, MARCM driver, heat-shock driver or at1 driver). The information of the driver used is also provided in the figure legends and Supplementary Table 1. To further improve clarity, we now also indicate the identity of the driver directly in Fig. 6, new Fig. 7, Fig. 8 (previously Fig. 7) and Fig. 9 (previously Fig. 8).

3. The use of statistics in this paper is definitely improvable: 1) the number of animals used for each experiment should be mentioned so that the reader can have an idea of the inter-individual variability for these phenotypes. 2) In those cases where the phenotype is just mild (Fig. 6E), a statistical test is necessary to understand the differences. 3) The phenotypes in Fig. 8 seem fully penetrant. Is that right? How many sensilla were analyzed?

RESPONSE: We provide separate response to these three important points:

1) Much of the data in this paper are more qualitative than quantitative in nature, where the phenotypes observed are consistent and reproducible in all animals examined. We now provide further information on the range of numbers of animals examined in the Methods. For all the quantitative data we provide the n values and have now appended the values to the heatmaps as well (Fig. 1e, Fig. 2e and Supplementary Fig. 1e) where there is more inter-individual differences due to stochastic developmental variation (the variation itself is reflected in the intensity of the heatmap plot).

2) We observed 33% doublets (with 2 copies of the driver) and 28% doublets (with an auto-amplifying UAS-GAL4 transgene) for *pnt* RNAi in Fig. 7b (previously Fig. 6e). As explained in the figure legend, the maximum percentage

of doublets in these experiments is in fact ~33% (indicated by the blue arrows in the figure) because the *at1* driver is only expressed in only a subset of the OSNs labelled by the *Or67d-CD8a:GFP* reporter. Thus, the *at1* driver induced *pnt* RNAi phenotype is nearly completely penetrant.

3) Our conclusion on Naa-to-Nab fate transformation is based on phenotypic examination of 20-50 sensilla (typically from more than 10 antennae) per experiment. In all cases, *pnt* RNAi has full penetrance but incomplete expressivity. All the individuals showed duplication of Nab neuron but the occurrence of Nab doublets may be more pronounced in certain sensillar type than others. We did not quantify the n or the doublet:singlet ratio for Fig. 9b-e (previously Fig. 8b-e) because these double RNA FISH experiments are simply meant to illustrate that the Nab doublet is located within the same sensillum as its sibling neuron (for example, Nba in Fig. 9c-d) or the supernumerary Nab is produced at the expense of the sibling Naa neuron in *ab1* sensillum (Fig. 9e). Quantitative information (n and ratio of Nab duplication for different sensillar types) is presented in Supplementary Fig. 5.

4. The effect of Pnt seems not too obvious when the *at1*-driver is used to drive RNAi (Fig. 6E). However, the authors use this driver to conclude that MAPK signal transduction is not involved in the Pnt phenotype. Is it possible that a similar phenotype could occur if this RNAi is driven by the more general drivers?

RESPONSE: As mentioned in the response above, the *at1* driver is capable of producing close to maximum level of *pnt* RNAi phenotype. However, the RNAi experiments of MAPK signal transduction pathway components were conducted with the constitutive driver (as with most of the experiments related to *pnt* RNAi). We have now indicated the driver used in Supplementary Fig. 5 for improved clarity.

Minor points

1. Fig. S1C is redundant.

RESPONSE: We acknowledge that Supplementary Fig. 1c is similar to Fig. 1b, but we hope to provide an intuitive graphical representation to direct the attention of the readers towards the PA region (in green), where *Bar-GAL4* and *ap-GAL4* are expressed (Supplementary Fig. 1d).

2. Showing those 19 drivers that gave inconsistent expression as supplementary material could be useful in future experiments.

RESPONSE: We believe that showing data on drivers with inconsistent expression patterns are more likely to mislead than to help the reader/future experimenters (especially when used without sufficient knowledge of their spatial and temporal expression dynamics). Nevertheless, we have now indicated the identity of these 19 drivers in Supplementary Table 6 for the interested reader (at

the very least to be regarded as drivers to *avoid* for immortalization experiments in the antennal disc).

3. When discussing a combination of drivers can define different territories, it should be remarked that this also includes *amos* and *ato*.

RESPONSE: We have amended the text accordingly.

4. Is the expression of the *GMR82D08* driver the same in males and females? This enhancer is endogenously located in the X chromosome and could be affected by sexual dimorphism, especially considering that *Or67d* is involved in courtship (Kurtovic et al., 2007).

RESPONSE: We examined both sexes in the course of our analysis of *GMR82D08-GAL4* but did not detect obvious dimorphic expression. We also note that *Or67d* OSN numbers are comparable in males and females (Grabe et al., Cell Reports 2016).

5. In the paragraph beginning in line 356, the specific drivers used for Gal4 expression should be named and discussed.

RESPONSE: We have now added the (simplified) genotypic description of the drivers in the main text for clarity. In addition, new panels have been added to new Supplementary Fig. 4 to detail the temporal expression pattern of the “constitutive” driver as it is, to our knowledge, used for the first time to drive expression in the antennal SOPs and OSNs.

6. In Fig. 6d, the analysis with the late Gal4 driver should also show both RNAi.

RESPONSE: The additional *pnt* RNAi line (JF02227) used with the constitutive drivers, as well as the MARCM clone for *pnt* mutant (new Fig. 7a and 7d) serve to rule out the possibility that the *Or67d* OSN duplication phenotype seen with KK100473 RNAi line (which was used in all other experiments) is an off-target effect. We believe that adding the panel of the second RNAi line (JF02227) for the late GAL4 driver (where a phenotype is *not* observed) would be redundant after we have established the KK100473 RNAi line is efficient and specific for inducing *pnt* RNAi.

7. There is a typo in line 508: “expression is consistent with an Naa...”

RESPONSE: We thank the reviewer for spotting the typo. The mistake has been rectified.

REVIEWERS' COMMENTS:

Reviewer #1 (Remarks to the Author):

My major concerns with previous manuscript were the embedded caveats of the immortalization labeling system, the minor inconsistency between their SOP/lineage results and previous reports from other groups, and the over-strong claim on no correlation between concentric arc organization in the periphery and the glomerular map in the central brain. The authors have addressed the majority of my previous concerns in their point-by-point responses and corrected/modified the text in response to some of these comments. (I did find it very hard and time-consuming to trace those modified paragraphs in the revision). Overall, the paper is significantly improved and the statement as it stands is much balanced. Therefore, I support the publication of this paper.

Reviewer #2 (Remarks to the Author):

The revised manuscript has satisfactorily addressed our previous concerns.